# Late Fusion Model for Emotion Recognition from Facial Expressions and Biosignals in a Dataset of Children with Autism Spectrum Disorder

**DOI:** 10.3390/s25247485

**Published:** 2025-12-09

**Authors:** Dominika Kiejdo, Monika Depka Prądzinska, Teresa Zawadzka

**Affiliations:** Faculty of Electronics, Telecommunications and Informatics, Gdańsk University of Technology, 80-233 Gdańsk, Poland; dominika.kiejdo@gmail.com (D.K.); moni.depka@gmail.com (M.D.P.)

**Keywords:** automatic emotion recognition, autistic children, facial emotion recognition, late fusion, neural network, heart rate, temperature, electrodermal activity

## Abstract

Children with autism spectrum disorder (ASD) often display atypical emotional expressions and physiological responses, making emotion recognition challenging. This study proposes a multimodal recognition model employing a late fusion framework combining facial expression with physiological measures: electrodermal activity (EDA), temperature (TEMP), and heart rate (HR). Emotional states are annotated using two complementary schemes derived from a shared set of labels. Three annotators provide one categorical Ekman emotion for each timestamp. From these annotations, a majority-vote label identifies the dominant emotion, while a proportional distribution reflects the likelihood of each emotion based on the relative frequency of the annotators’ selections. Separate machine learning models are trained for each modality and for each annotation scheme, and their outputs are integrated through decision-level fusion. A distinct decision-level fusion model is constructed for each annotation scheme, ensuring that both the categorical and likelihood-based representations are optimally combined. The experiments on the EMBOA dataset, collected within the project “Affective loop in Socially Assistive Robotics as an intervention tool for children with autism”, show that the late fusion model achieves higher accuracy and robustness than unimodal baselines. The system attains an accuracy of 68% for categorical emotion classification and 78% under the likelihood-estimation scheme. The results obtained, although lower than those reported in other studies, suggest that further research into emotion recognition in autistic children using other fusions is warranted, even in the case of datasets with a significant number of missing values and low sample representation for certain emotions.

## 1. Introduction

Recognizing human emotions automatically is an increasingly important area in affective computing, with potential applications in healthcare, education, and human–computer interaction. In particular, emotion recognition for children with autism spectrum disorder (ASD) is a growing field of interest, as these children often experience difficulties in expressing and interpreting emotions [1]. Accurate recognition of their emotional states can help support therapeutic processes, improve communication, and enable adaptive educational environments. The EMBOA project entitled “Affective Loop in Socially Assistive Robotics as an intervention tool for children with autism” [2] focused on developing and evaluating tools that support the recognition and understanding of emotions in children with autism (https://emboa.eu/). Although the project has concluded, the dataset collected within its framework provides valuable resources for further research. Building on this foundation, the present study develops an automatic emotion recognition model based on a late-fusion strategy that integrates facial expression data with physiological signals. Unlike unimodal systems, which rely on a single type of data, a multimodal approach was used, incorporating facial features extracted via FaceReader software, electrodermal activity (EDA), temperature (TEMP), and heart rate (HR) signals.

The goal of this research is to investigate whether it is possible to construct a reasonable late fusion model for emotion recognition when working with a highly challenging dataset that is greatly imbalanced [2]. Late fusion was selected as the first choice due to the recording time, during which it was not possible to detect faces in order to extract the features needed for automatic facial emotion recognition. Therefore, a late fusion model was developed. It integrates outputs from separate unimodal models into a single, robust emotion recognition system. The dataset was annotated by three independent raters using the BORIS software, version 8.21.3 (https://www.boris.unito.it/). Each annotator assigned exactly one categorical emotion for each timestamp, selecting from the six Ekman basic emotions. From this single annotation process, two labeling strategies were derived: a binary emotion classification approach using majority voting to obtain a final label, and percentage-based emotion representation in which all three annotations were preserved and converted into proportional distributions, enabling the capture of ambiguity. The dataset was labeled as part of an engineering project [3].

By applying late fusion, this research aims to assess whether combining information from multiple unimodal models can mitigate the limitations posed by the dataset’s imbalance and ultimately improve the robustness of emotion recognition under such restrictive conditions.

## 2. Related Work

Emotion recognition from physiological signals and facial expressions has attracted considerable attention in affective computing. Early efforts often focused on unimodal pipelines (e.g., EDA-only or HR-only) [4,5,6,7], while more recent approaches increasingly explore multimodal fusion to leverage complementary cues [8,9,10]. First, a summary of representative unimodal studies is presented, focusing on electrodermal activity, heart rate, temperature, and facial emotion recognition, followed by a review of multimodal fusion strategies that motivate our late-fusion design and their application to emotion recognition in autistic children. A compact overview of related works is presented in Table 1. It summarizes representative studies on unimodal and multimodal emotion recognition.

The works are grouped by signal type (EDA, HR, TEMP, Face). It is particularly observed that multimodal fusion studies have increased in contexts involving children with ASD. However, despite growing interest, the number of studies specifically targeting emotion recognition in children with ASD remains very limited. Most existing pipelines are developed for neurotypical populations, and their direct applicability to ASD contexts is often unclear. This scarcity is partly due to the challenges of labeling such datasets, as emotional expressions may be atypical, ambiguous, or inconsistent across annotators [11].

Both classical baselines (e.g., [12,13]) and recent deep learning and fusion approaches (e.g., [14,15]) are included to illustrate the evolution of methods. The selected studies were chosen because they either focus directly on ASD populations (e.g., EMBOA and in-house ASD datasets) or present multimodal emotion-recognition pipelines. In addition, representative unimodal methods—such as those based on electrodermal activity (EDA), heart rate (HR), skin temperature, or facial expressions—were included to highlight the strengths and limitations of single-modality approaches relative to multimodal fusion.

**Table 1 sensors-25-07485-t001:** Compact summary of related works, grouped by modality/fusion type. Source column cites the bibliography entries.

Source	Type of Model	Year	Modalities	Age	ASD	Emotions	Dataset
EDA-based approaches							
[12]	CNN (ResNet), RNN (LSTM), Hybrid (ResNet-LSTM)	2020	EDA	Adults	No	valence and arousal	MAHNOB
[16]	FCN	2023	EDA	N/A	No	non-active vs. active	In-house
HR-based approaches							
[13]	KNN, RF, DT, GBDT, AdaBoost	2020	HR	Young adults	No	happy, sad, neutral	In-house
[17]	SVM/KNN/RF	2023	HR	Children	Yes	positive, negative, neutral	In-house
Face-based approaches							
[3]	CNN	2023	Face	Children	Yes	joy, sadness, anger, fear, surprise, neutral	EMBOA
[18]	CNN/VGG16	2022	Face	Adults	No	anger, disgust, fear, happiness, sadness, surprise, neutral	FER-2013
TEMP-based approaches							
[19]	GNB/SVM (RBF)/MLP	2025	TEMP	Adults	No	joy, love, fear, sadness	In-house
Multimodal fusion approaches							
[14]	Two-stage cascaded multimodal (audio and video)	2021	Speech + Face	Children	Yes	positive, neutral, negative	ASD-Affect
[15]	Siamese + LSTM (late fusion)	2025	BVP + EDA	Adults	No	neutral, stress, amusement, meditation	WESAD
[20]	CNN + GRU	2025	TEMP+ HRV	Adults	No	anger, disgust, fear, happiness, sadness, surprise	In-house

### 2.1. Emotion Recognition from Electrodermal Activity

Electrodermal activity (EDA) has long been recognized as a reliable surrogate of sympathetic arousal, since sweat glands are exclusively innervated by the sympathetic nervous system, which is directly involved in emotion regulation [21]. As such, EDA captures changes in physiological arousal that can be leveraged for stress and affect recognition.

Building on these foundations, ref. [12] conducted a systematic exploration of deep neural network families for EDA-based emotion recognition, comparing CNN, RNN, and hybrid architectures and highlighting design choices that affect generalization. Similarly, ref. [16] applied a fully convolutional network (FCN) to EDA signals to distinguish between neutral and active/stress states. While these EDA-only systems are promising, they primarily capture general arousal rather than differentiating specific emotions.

### 2.2. Emotion Recognition from Heart Rate

Heart rate (HR) is a widely used index of autonomic nervous system activity and plays a central role in emotion regulation. Polyvagal theory suggests that variability in heart rate provides a sensitive marker of one’s ability to engage in social communication and recognize emotions, linking parasympathetic regulation to adaptive social cognition [22]. HR has been shown to positively correlate with performance on emotion recognition tasks, highlighting its potential as a biological marker of affective processing.

Recent studies have explored HR for automatic emotion recognition using wearable devices. Lin Shu [13] employed wrist-worn smart bracelets to capture HR responses to emotional video stimuli in young adults, demonstrating that normalized HR features combined with gradient boosting classifiers could distinguish happy, sad, and neutral states with high accuracy. Similarly, Ali et al. [17] investigated HR-based emotion recognition in children with autism spectrum disorder, using SVM, KNN, and RF classifiers to distinguish between positive, negative, and neutral states. Their results highlight both the potential and the challenges of applying HR-based methods in pediatric and clinical populations.

Nevertheless, HR-only systems face inherent challenges. HR is influenced not only by affective arousal but also by non-emotional factors such as cognitive load, posture, or physical activity, making it difficult to disambiguate emotion states [5].

### 2.3. Emotion Recognition from Facial Expressions

Facial expression analysis remains one of the most widely explored modalities in affective computing due to its direct relation to observable emotional cues. The rise of deep learning has largely replaced manual feature extraction with convolutional neural networks (CNNs) capable of learning hierarchical visual features end-to-end [6]. For example, ref. [3] evaluated CNN-based facial emotion recognition models on the EMBOA dataset of autistic children and reported limited accuracy, illustrating the difficulty of generalizing models trained on neurotypical populations. In contrast, CNN/VGG16 models trained on adult datasets such as FER-2013 [18] achieve much higher accuracy, demonstrating the difference in model performance between neurotypical adults and children with ASD. Facial cues show both potential and limitations: they are highly informative in adults under controlled conditions, but their reliability decreases in populations such as children with ASD, where expressions may be atypical or less synchronized with internal states.

### 2.4. Emotion Recognition from Temperature

Body temperature variations have recently emerged as informative physiological correlates of emotional state due to their close coupling with autonomic activity and vascular regulation [19,20]. Unlike EDA or HR, which capture electrical or rhythmic activity, temperature reflects peripheral vasoconstriction and thermoregulation processes, providing complementary insight into stress and affective valence. Classical work primarily relied on peripheral skin temperature (e.g., wrist, hand, or facial thermography) to assess stress and arousal. More recently, ref. [19] introduced a novel approach using bilateral tympanic membrane temperature (TMT) measured through custom earphones. They demonstrated that right–left TMT asymmetry encodes emotion-related hemispheric activity. This study established the feasibility of earphone-based thermal sensing for continuous emotion recognition. Complementarily, ref. [20] developed a hybrid CNN–GRU model that combines skin temperature and heart rate variability (HRV). Although bimodal in nature, the architecture is dominated by thermal features and demonstrates the power of temporal modeling for physiological emotion recognition. Their system highlighted the potential of temperature dynamics as a discriminative affective signal.

### 2.5. Multimodal Fusion Approaches

To overcome the limitations of unimodal pipelines, multimodal fusion has emerged as a dominant trend. Representative work includes [15], which applies a Siamese network with late fusion of EDA and blood volume pulse (BVP) on the WESAD dataset, reporting superior robustness compared to single-modality baselines. Late fusion can be particularly advantageous when modalities differ in noise profiles or sampling rates, since decision-level integration reduces the risk of error propagation from any single stream.

Additional evidence comes from autism-centered contexts: the two-stage multimodal framework proposed by [14] leverages speech cues to detect negative affect before applying frame-wise facial expression recognition to differentiate between positive and neutral states. This cascaded approach effectively combines modalities without requiring strict temporal alignment, which is especially useful in naturalistic ASD therapy recordings where audio and video quality vary significantly. More broadly, recent reviews highlight that multimodal pipelines consistently outperform unimodal ones across domains, particularly in ASD-related applications where expressive and physiological cues complement each other [8,9,10,23].

### 2.6. Emotion Recognition in Autistic Children

Compared to general emotion recognition, work focusing on autistic children is more recent and faces distinct challenges. Individuals with ASD often display atypical and less synchronized facial expressions, making models trained on neurotypical populations less effective [1]. The EMBOA project explicitly addressed this gap by collecting multimodal data from children with autism for affective computing and therapeutic support. Within this context, several works explore unimodal and multimodal solutions: Ref. [17] trained SVMs on HR data from children with ASD; Ref. [3] analyzed facial expressions in children, reporting limitations of unimodal face-based models; and Ref. [14] proposed a two-stage multimodal framework combining speech-based detection of negative affect with frame-wise facial expression recognition for distinguishing positive and neutral states. These findings collectively emphasize that multimodal integration offers significant advantages in ASD emotion recognition, where single modalities (e.g., face or HR) are often insufficient due to atypical expression patterns and data variability.

Beyond computational approaches, meta-analytic evidence also underscores the depth of emotion processing impairments in ASD. A large-scale review of 148 studies found significant and nonselective deficits in recognizing all basic facial emotions, with impairments generally more severe in ASD than in other clinical conditions [24]. These impairments were not limited to facial attributes but extended to emotion recognition across modalities, suggesting a generalized processing deficit moderated by task complexity and stimulus characteristics. At the neural level, a recent meta-analysis of functional neuroimaging studies revealed abnormal activation patterns in ASD during negative emotion processing, including hypoactivation in limbic and fusiform regions and hyperactivation in temporal gyri [25]. These findings link behavioral deficits to atypical engagement of the mirror neuron and limbic systems and show correlations between abnormal activation and the severity of social–communication difficulties. Together, behavioral and neuroimaging evidence suggest that impaired emotional processing in ASD is both multimodal and neurobiologically grounded, further motivating multimodal affect recognition systems tailored to this population.

## 3. EMBOA Dataset

The dataset used in this study originates from the EMBOA project, entitled “Affective loop in Socially Assistive Robotics as an intervention tool for children with autism”, a research and educational initiative conducted between 2019 and 2022 under the Erasmus+ Strategic Partnerships Programme for Higher Education [2]. The project offers a comprehensive collection of multimodal recordings, aimed at supporting emotion recognition in children with autism. The dataset includes video recordings, audio, eye-tracking data, and physiological signals gathered via wearable wristbands. The physiological parameters recorded are accelerometer data (ACC), photoplethysmographic signal (BVP), electrodermal activity (EDA), heart rate (HR), inter-beat intervals (IBI), and skin temperature (TEMP).

Despite the broad range of available modalities, this research focuses solely on facial expressions detected using the FaceReader software, along with three physiological signals; heart rate (HR), temperature (TEMP), and electrodermal activity (EDA); due to their established relevance in affective computing. Emotional labels in the dataset were created using two annotation methods based on BORIS software. The available emotion categories were contempt, happy, sad, scared, disgusted, surprised, and angry. If, for a given second, none of the seven emotions were selected, BORIS recorded that moment with no assigned emotion. In our work, these instances were designated as “other”, indicating that the child’s expression did not match any of the defined categories. These categories are derived from Paul Ekman’s theory of basic emotions, which identifies universal facially expressed emotions such as happiness, sadness, anger, fear, surprise, and disgust, with contempt later added to reflect social and moral judgment [26]. The annotation was carried out independently by three annotators (female students from Gdańsk University of Technology). Annotators based their decisions primarily on facial expressions and body movements, but they also considered the context of each scenario and, when understandable, elements of speech and vocal tone [3]. Two labeling strategies were constructed:Method I assigns a single dominant emotion for each one-second interval. This is determined by collecting the three annotations and selecting the most frequently chosen label. The available emotion categories were contempt, happy, sad, scared, disgusted, surprised, angry, or *other* (used when none of these seven emotions were selected). If all three annotators selected different emotions (i.e., no majority label was present), the interval was assigned to the *other* category, as no dominant emotion could be determined.Method II computes percentage-based emotion distributions by averaging three independent categorical annotations across seven basic emotions: contempt, happy, sad, scared, disgusted, surprised, and angry. Each annotator provides one categorical label per second; these labels are converted into a normalized distribution, yielding a probabilistic representation of affective states. If an interval was labeled as *other*, this means that none of the seven emotions was annotated for that second. If all three annotators selected different emotions, each of the three selected categories received a probability of 33.33%. If an interval was labeled as *other*, this means that none of the seven emotions was annotated for that second, resulting in a zero vector.

For the purposes of further analysis, the category *contempt* was discarded, as it does not have a direct correspondence with the emotions detected by FaceReader [3]. Consequently, intervals labeled as *contempt* were treated in the same way as *other*.

### 3.1. Participants

This study involved children aged 0 to 12 years, including both females and males. Participants were recruited and studied at several research centers: Gdańsk University of Technology (Poland), Istanbul Technical University (Turkey), Yeditepe University (Turkey), and the Macedonian Association for Applied Psychology (North Macedonia). Gdańsk University of Technology coordinated the project, while the University of Hertfordshire was responsible for developing the humanoid robot *Kaspar*, used in the interventions [27].

### 3.2. Experimental Scenarios

Each session with a child consisted of structured scenarios designed to elicit both social and emotional reactions during interaction with Kaspar. The core scenarios, shared across all centers, included:Singing a song (e.g., *Happy and you know it* or localized equivalents such as *Panie Janie*).Emotion scenarios, where Kaspar displayed a basic emotion (happy, sad, surprised, hiding), and the child was asked to mimic it.Animal scenario, in which Kaspar asked about animal sounds (e.g., “What does a cat say?”).Body parts scenario, requiring the child to point to named body parts.Movement imitation, where the child repeated Kaspar’s gestures (e.g., raising a hand).Sound imitation, where the child repeated simple sounds or syllables spoken by Kaspar.

### 3.3. Data Collection and Quality Issues

The dataset included a total of 108 video recordings, of which 99 were analyzed after removing duplicates and 1 corrupted file. The recordings correspond to approximately 16 h and 50 min of interaction time. The majority of the data came from the Macedonian center (72%). Despite minor issues—such as duplicate recordings or unreadable video files—the dataset retained sufficient breadth and quality to ensure reliability of the analyses [3].

### 3.4. Data Preparation

The raw EMBOA recordings required preprocessing to ensure consistent sampling, cross-modal alignment, and suitability for sequence models targeting emotion detection. The pipeline was structured around three stages: feature extraction, temporal synchronization and standardization, and class balancing.

#### 3.4.1. Feature Extraction

A multimodal setup was used combining facial expression analysis with peripheral physiology. For the visual stream, the child’s face was analyzed using FaceReader [28], which performs frame-by-frame emotion classification (following the video frame rate) but outputs an event-based log containing only dominant-emotion change points [3]. Because this log is irregularly sampled, it was converted into a regular 1 Hz time series. Specifically, This was performed by assigning each second the most recent preceding label (forward-fill between change points), resulting in a continuous per-second sequence of categorical emotion labels aligned with the physiological signals.

Physiological signals comprised electrodermal activity (EDA), heart rate (HR), and temperature (TEMP) acquired via wristband sensors. EDA and TEMP were sampled at 4 Hz and HR at 1 Hz. To align these signals, a 1 Hz representation was generated by averaging the 1-second bins of EDA and TEMP and retaining the native 1 Hz HR values.

This produced quad-modal, co-sampled sequences at 1 Hz containing EDA, HR, TEMP, and the face-derived emotion labels. To prepare the synchronized data for sequence-based modeling, fixed-length windows were extracted from the 1 Hz aligned streams.

A 4-second window length was selected. Prior work in physiological emotion recognition has shown that window sizes in the range of 3–8 s provide a suitable trade-off between temporal sensitivity and signal stability [29,30]. A 2-s stride (50% overlap) was used to increase the number of available samples while preserving temporal continuity without introducing excessive redundancy.

#### 3.4.2. Data Standardization

Numeric biosignals (EDA, HR, TEMP) were standardized per subject using Z-score normalization, following standard statistical practice [31],(1)$$Z=\frac{X-\mu }{\sigma }$$
where *X* represents the raw signal value, μ is the mean of the signal for the corresponding subject, and σ is the standard deviation of that signal. This standardization mitigates scale differences across sessions. Recording timestamps, video durations, and start offsets were extracted and cross-checked using FFmpeg utilities [32]. Sensor start times (EDA/HR/TEMP) were read from wristband metadata to anchor all streams on a common timeline.

In parallel, emotional annotations from the EMBOA protocol (Method I and Method II) were aligned to the same timeline. Method I provides a single categorical label per second based on majority vote across three annotators, whereas Method II represents the proportional likelihood of each emotion according to the annotators’ distributions. These two complementary annotation schemes were temporally synchronized with the biosignals and FaceReader stream.

The resulting feature space consists of synchronized, standardized 1 Hz sequences of EDA, HR, and TEMP, together with three aligned label streams: (i) the 1 Hz FaceReader-derived dominant emotion, (ii) the categorical majority-vote label (Method I), and (iii) the proportional likelihood distribution (Method II).

#### 3.4.3. Data Balancing

The raw label distribution was highly skewed, with no emotion labeled *other* and *happy* dominating, and minority classes —particularly *disgusted*, *angry*, and *surprised* —appeared only rarely. To reduce majority-class bias, two complementary strategies were applied at the sequence level:**Sequence filtering**For Method I, sequences were excluded that contained only *other* and/or *happy* throughout their duration. Sequences were retained if they included at least one frame labeled with any minority emotion (i.e., any label other than *other*/*happy*), or if the *happy* column itself had fewer than two but at least one non-zero entry.For Method II, only those segments were excluded that contained exclusively *other*, as well as segments annotated only with *other* and *happy* across their full duration. Segments labeled solely with *happy* were not excluded.**Per-class balancing**Among the retained frames containing minority emotions, resampling was applied only to the training set to equalize class frequencies and increase the amount of minority-class data. Frames whose dominant emotion was *disgusted*, *surprised*, or *angry* were up-sampled.Each minority-class frame (1 Hz) was expanded into four frames using the original physiological data recorded at 4 Hz. For every such 1-second interval, the up-sampled sample was constructed as follows:the four native 4 Hz TEMP and EDA samples corresponding to the same interval (Section 3.4.1),the HR value repeated across the four frames (native 1 Hz sampling rate),the FaceReader-derived emotion label copied to all four frames.This produced one additional four-frame sample for each original minority-class frame. All artificially upsampled samples were added exclusively to the training set. The validation and test sets were not resampled or otherwise augmented.

The effect of these steps is illustrated for Method I in Figure 1 and Figure 2, which show only the training dataset. Before balancing, the training dataset was strongly dominated by *happy* and *other* labels. After balancing, the distribution across all emotion categories becomes more uniform, ensuring that minority classes are better represented during model training.

The corresponding effect for Method II on the training dataset is shown in Figure 3 and Figure 4. Each Method II chart reports percentages within the training set; for every emotion, the three bars represent the mapped intensity (or strength) levels, approximately 33.33%, 66.66%, and 100%.

#### 3.4.4. Dataset Split

A subject-disjoint split of 60%/20%/20% (train/val/test) was performed before upsampling (Section 3.4.3). This ensures that no synthetic or duplicated samples leak into the validation or test sets. The validation set was used to tune decision-level fusion weights in our late-fusion experiments, following best practices for multimodal late fusion, where per-modality hypotheses are combined on a held-out set [33]. The test set remained untouched for final performance reporting.

## 4. Research Method

The methodological framework adopted in this study is based on the EMBOA dataset. The framework employs a dual analysis of emotional states, designed to capture the inherent complexity and ambiguity of affective expression in children with autism spectrum disorder (ASD) [3]. The first analytical path aims to determine the dominant emotion. The second path estimates the distribution across all emotions, enabling representation of blended or overlapping emotional states. It can also serve as an alternative way to determine the dominant emotion by selecting the emotion with the highest intensity. This dual approach facilitates both discrete emotion classification and graded emotional profiling, addressing the challenges posed by the often subtle, atypical, or ambiguous emotional expressions characteristic of children with ASD.

A late fusion strategy was employed to integrate the multimodal models’ outputs. Separate models were trained independently for each modality, with their outputs combined at the decision model level. Although late fusion has known limitations—particularly when unimodal predictions conflict—it also offers several practical advantages relevant to our experimental setting. First, it allows each modality to be modeled independently, which is consistent with established practice in multimodal affect recognition, where separate classifiers are trained and their outputs combined at the decision stage. This design provides flexibility: models for individual modalities can be adapted or replaced without retraining the entire system. Second, late fusion avoids the high-dimensional feature concatenation inherent to early fusion, which can require substantially larger datasets to ensure stable learning [34].

The overall pipeline, including feature extraction, unimodal modeling, and late fusion at decision level, is illustrated in Figure 5. The framework consists of four parallel streams for electrodermal activity, heart rate, temperature, and facial video, each undergoing feature extraction (Section 3.4.1), and data standardization and alignment (Section 3.4.2) are followed by data balancing, where per-class balancing was applied explicitly to the training dataset only (Section 3.4.3). Following preprocessing, unimodal models are trained for each modality (Section 5.1.1), and their outputs are integrated through a decision-level late fusion strategy (Section 5.1.2). Although FaceReader natively provides dominant-emotion labels, its output is additionally transformed for Method II to generate a proportional emotion-distribution vector rather than a single categorical label. This design leverages both physiological and behavioral signals to yield robust emotion recognition outcomes.

## 5. Experiments Design

This section presents the design of the experiments conducted to evaluate the proposed emotion recognition framework. Two modeling approaches were developed based on the format of emotion labels: one utilizing categorical emotion labels (Method I), and the percentage-based representations (Method II). Both approaches incorporated multimodal data sources, including electrodermal activity (EDA), heart rate (HR) and temperature (TEMP), and facial expression features extracted via FaceReader. A late fusion strategy was employed to integrate information from these modalities at the decision level.

### 5.1. Models’ Architectures

Given that physiological signals such as EDA, HR, and TEMP are inherently sequential, recurrent neural networks were adopted to capture temporal dependencies within the data. In particular, Bidirectional Long Short-Term Memory (BiLSTM) layers were selected as the core component, as they allow the model to exploit both past and future context within a sequence, which is critical for detecting short-term emotional fluctuations.

For facial expression features extracted with FaceReader, which provide frame-level emotion probabilities, sequences were constructed over fixed temporal windows to preserve dynamic changes in facial behavior. This ensured that all modalities were represented as time-series inputs, enabling consistent processing across channels.

The following subsections describe the architectures used for individual modality models as well as the late fusion strategies for combining modality-specific predictions.

#### 5.1.1. Individual Modality Models

Separate models were constructed for each physiological modality (EDA, HR, TEMP) and for each annotation scheme, each adopting a deep recurrent neural network architecture with a Bidirectional Long Short-Term Memory (BiLSTM) layer, followed by fully connected dense layers. For the behavioral modality, facial embeddings and emotion features were extracted using FaceReader (FR), a commercial software tool for automated facial expression analysis [28]. For the categorical annotation scheme (Method I), the FaceReader stream provided a per-second dominant-emotion label, which was used directly as the behavioral input to the decision model. For the percentage-based annotation scheme (Method II), the FaceReader output was further transformed using the neural modeling procedure. All models were trained with a batch size of 32 and input sequence lengths of 4 time steps.

Method I (Categorical Labels): These models were designed to classify emotions into discrete categories. Each network comprised a BiLSTM layer with *128 units*, followed by two dense layers with 64 and 32 units, respectively, and a final softmax-activated output layer. A custom masked categorical cross-entropy loss was implemented to ignore samples with unknown labels (encoded as vectors of zeros), ensuring robust learning. Frames labeled as *Other* were retained in the input sequences to preserve their temporal structure but were excluded from loss computation through the masking mechanism. This ensured that the network learned only from frames corresponding to one of the six target Ekman emotions, without treating *Other* as a valid class or propagating gradients from unlabeled moments. Early stopping was employed with a patience of 10 epochs to prevent overfitting.Method II (Percentage-Based Labels): Each modality is processed using its own unimodal model. For physiological signals—EDA, TEMP, and HR—the model takes a short sequence of four consecutive signal values. These models start with a bidirectional LSTM layer containing 128 units, which allows the network to learn temporal patterns in both forward and backward directions. The output of this recurrent layer is then passed through two fully connected layers with 128 and 64 ReLU units, which further transform the learned transform representation. Finally, the model outputs six emotion values through a sigmoid layer, producing continuous estimates for each emotion. The FaceReader modality uses a simpler unimodal model because the input already consists of six emotion values per timestep. Instead of using recurrent processing, this model directly applies a fully connected layer with 64 ReLU units to each timestep of the four-frame sequence, followed by a sigmoid output layer that predicts six emotion probabilities. All unimodal networks are trained separately using the Adam optimizer and masked mean squared error loss that ignores missing labels. Training uses a batch size of 32 and runs for 40 epochs for each modality.

#### 5.1.2. Late Fusion Models

A late fusion approach was employed to integrate modality-specific predictions. Importantly, the final fusion models for both methods were trained exclusively on the predictions generated by the unimodal classifiers. Intervals labeled as *other* (Method I) or frames represented by all-zero vectors indicating no annotated emotion (Method II) were excluded from training and validation through masking and were never treated as a separate output class. Consequently, the system always outputs one of the six emotions and cannot explicitly assign an input to an “other” or “unknown” category. In practice, this means that the models do not abstain from making a decision; instead, they always select the most probable of the six emotions.

Method I: Logistic Regression Fusion (Categorical Labels)For categorical emotion labels, late fusion was applied to integrate predictions from multiple modality-specific models, including EDA, HR, TEMP, and FaceReader. First, out-of-sample predictions were obtained from each base model.The vectors from all modalities were then concatenated into a single feature representation, resulting in a fused vector of dimension 6×4=24 per valid frame. These fused features served as input to a multinomial logistic regression classifier (scikit-learn implementation), trained with balanced class weights.Method II: Weighted Late Fusion Model (Distribution)For Method II, a late fusion was employed to integrate predictions from unimodal networks, enabling the system to learn temporal dependencies across modalities while automatically adjusting the relative contribution of each signal. The fused input consists of the concatenated outputs from the four unimodal models (EDA, HR, TEMP, FaceReader) across short temporal dynamics in both forward and backward directions. The recurrent output is then passed through a fully connected layer with 64 ReLU units to model interactions between modalities. Finally, a sigmoid-actived output layer produces continuous probabilities for six emotions, providing the final multilabel emotion predictions. The model is trained using a masked mean squared error loss and optimized with Adam algorithm.

### 5.2. Model Training and Settings

All models were trained and evaluated using fixed-length sequences extracted from the synchronized multimodal recordings. Each sequence contained time-aligned physiological signals (EDA, HR, TEMP) and facial expression features and was annotated with either categorical labels (Method I) or percentage-based emotion intensities (Method II). Frames encoded as all-zero vectors—corresponding to the Other category, where none of the six Ekman emotions was active—were treated as missing labels. These frames were retained within the temporal sequences but excluded from backpropagation using custom masking functions applied to the loss computation. To ensure methodological rigor and to avoid any leakage between training and evaluation stages, a subject-disjoint data partitioning strategy was adopted. The dataset was first divided into fixed splits of 60% for training, 20% for validation, and 20% for testing, and these splits remained identical for all models and both annotation schemes. Crucially, balancing procedures were applied exclusively to the 60% training partition. Neither the validation nor the test sets were resampled, augmented, or modified in any way. This guaranteed that all reported performance values reflect generalization to unseen and naturally distributed data. Model training followed a sequential, two-stage pipeline. First, unimodal models for each modality (EDA, HR, TEMP, FaceReader) were trained independently on the balanced training subset. After convergence, these unimodal models were used to generate predictions for the training sequences only, which served as inputs for training the late-fusion classifier. During evaluation, unimodal predictions were computed strictly from the test sequences and forwarded to the late-fusion model, ensuring a complete separation between training-generated and evaluation-generated predictions.

Evaluation was conducted separately for Method I and Method II. For Method I, balanced accuracy was adopted as the primary metric due to the pronounced class imbalance in the categorical labels. For Method II, performance was evaluated using the mean squared error (MSE) computed over the six continuous emotion-intensity channels. Following established practice in continuous affect estimation [35,36], the MSE was computed independently for each emotion, while frames with all-zero annotations—providing no valid supervisory signal—were excluded through a binary mask. A single overall score was then obtained by taking the unweighted mean of the six per-emotion MSE values.

All models used consistent input sequence lengths and were trained with early stopping based on validation loss. Method I models were trained for up to 100 epochs. For Method II, unimodal models were trained for 40 epochs, while the late-fusion classifier was trained for 50 epochs. This configuration ensured stable convergence while preventing overfitting across both annotation paradigms.

## 6. Performance Evaluations

Two final models were evaluated on a six-class task: Method I (categorical labels) and Method II (percentage-based labels). Each method was assessed using appropriate methods for its labeling scheme.

### 6.1. Method I: Evaluation of Late Fusion Model

The late fusion model of Method I was evaluated on the six-class emotion recognition task using the EMBOA dataset. Individual modality models (EDA, HR, TEMP, and FaceReader) produced probability distributions over the six emotion classes, which were combined using a decision-level fusion mechanism.

For evaluation, the same feature construction process was applied to the test set. The trained logistic regression model produced the final fused predictions. Performance was assessed using standard multi-class classification metrics, including accuracy, micro- and macro-averaged precision, recall, F1-scores, Hamming loss, and balanced accuracy [37]. These metrics follow established recommendations for evaluating classifiers under class imbalance.

#### 6.1.1. Unimodal Performance for Method I

Before evaluating fusion strategies, we first assessed the predictive performance of individual modality-specific models (EDA, HR, TEMP, FaceReader) trained with categorical labels. Each model followed the architecture described in Section 5.1.1.

The unimodal results, summarized in Table 2, reveal substantial variability across physiological and visual modalities. Among physiological signals, the EDA model achieved the highest performance, the highest performance, with an accuracy of 0.70 and a balanced accuracy of 0.26, suggesting that electrodermal activity is particularly informative for distinguishing arousal-related emotional states. HR and TEMP performed less effectively, with balanced accuracies of 0.16 and 0.24, respectively. The FaceReader model, based on visual features, achieved an accuracy of 0.40 and a balanced accuracy of 0.18. While capable of detecting high-frequency classes such as *Happy*, it struggled to differentiate between less represented expressions, which is consistent with challenges reported in the literature regarding facial emotion analysis [38].

Overall, all unimodal models exhibited low macro-F1 scores, reflecting weak recognition of minority classes (*surprised*, *scared*, *angry*). These limitations motivated the development of multimodal late fusion to exploit complementary information across signals.

As shown in Table 2, no single modality provided sufficient performance for robust multi-class emotion recognition. While EDA provided the strongest unimodal signal, all models suffered from low balanced accuracy and macro-F1, indicating strong biases toward majority classes and poor recognition of minority emotions. These findings support the use of multimodal fusion, where modalities can be leveraged to improve balanced classification performance.

#### 6.1.2. Late Fusion Performance for Method I

The metrics in Table 3 provide a detailed view of the model’s behavior across classes. The overall accuracy and micro-averaged F1-score of 0.68 indicate that the fused model effectively captures the dominant patterns present in the dataset. The balanced accuracy of 0.64 shows that performance is not uniform across classes, which is an expected outcome given the inherent class imbalance in the EMBOA dataset.

Although the balanced accuracy remains lower than the overall accuracy, it is nevertheless substantially higher than the balanced accuracies achieved by any unimodal model. This demonstrates that the fusion model successfully integrates complementary cues across modalities, improving the recognition of minority emotions and mitigating the strong class biases observed in unimodal classifiers.

The macro-recall of 0.64 shows that the model is able to detect minority classes more often than it correctly classifies them (lower macro-precision), suggesting that false positives are more common than false negatives for these classes. This behavior is also reflected in the Hamming loss of 0.32, which quantifies misclassification on a per-instance, per-label basis. A loss of this magnitude indicates that nearly one-third of all label assignments deviate from the ground truth, consistent with the observed confusion patterns across the more ambiguous or underrepresented emotions.

The confusion matrix in Figure 6 illustrates the percentage distribution of predictions for each true emotion class. The model performs particularly well for the *sad* (76.0%) and *scared* (78.0%) classes, achieving the highest true positive rates. The *happy* class also shows strong recognition (65.7%), although some confusion is observed with *surprised*, *angry*, and *disgusted*.

For minority classes, performance remains more variable. The *disgusted* class is correctly identified in 66.7% of cases but is frequently misclassified as *happy* (33.3%). The *surprised* class is correctly detected in 50.0% of instances, with errors spread primarily across *happy*, *scared*, and *disgusted*. The *angry* class shows recognition of (53.8%) and is often confused with *happy* (30.8%).

Overall, Method I demonstrates that late fusion with a logistic regression meta-model effectively integrates multimodal predictions, improving overall accuracy while mitigating some weaknesses of unimodal systems.

### 6.2. Method II: Evaluation of Late Fusion Model with Multi-Label Annotations

Method II applies late fusion to continous, multi-label emotion estimation. These ouputs are combined using a regression-based fusion model. Performance is evaluated using mean square error (MSE) and mean absolute error (MAE) for individual emotions, while cosine similarity is calculated only on the overall predictions, capturing the alignment between predicted and true emotional patterns.

#### 6.2.1. Unimodal Performance for Method II

Unimodal model performance was assessed using mean squared error (MAE) across electrodermal activity (EDA), heart rate (HR), temperature (Temp), and facial expressions (FR). The results presented in Table 4 provide insight into the realtive predictive value of each physiological modality for emotion recognition in autistic children. Across all emotions, the EDA model achieved the lowest MSE (0.133), indicating that electrodermal activity offered the most informative unimodal signal for this task. This pattern reflected consistently across several emotions. Notably, the EDA model produced the lowest error for Sad (0.085), Scared (0.097), and Happy (0.323). It also performed competively on Disgusted (0.082), matching the HR model and slightly outperforming both the Temperature and FR models. The Temperature model showed moderate overall performance (0.141), with particularly low errors for Disgusted (0.081) and Surprised (0.098). The HR model obtained a slightly higher overall error (0.147) and generally performed below the EDA and Temperature models, although it remained competitive for Surprised (0.099) and Angry (0.105). The FR model recorded the highest overall error (0.152), with consistently weaker results across most emotions, reflecting the reduced reliability of facial cues in this population. Overall, EDA emerged as the strongest unimodal modality, followed by Temperature and HR, while FR contributed the least predictive value.

#### 6.2.2. Late Fusion Performance for Method II

The late fusion model demonstrated improved prediction accuracy compared to the unimodal approaches, achieving an overall MSE of 0.1133 and an overall MAE of 0.2910 (Table 5). These results indicate that combining multiple physiological signals provides a more reliable representation of emotional states than any single modality alone. The inclusion of cosine similarity further supports this improvement by capturing the directional agreement between predicted and ground-truth emotional patterns. Per-emotion results further highlight this improvement. The lowest errors are observed for Scared and Sad, suggesting that these emotions benefit most from multimodal integration. Cosine similarity of these classes (0.868 and 0.804, respectively) also indicates strong alignment between predicted and true emotional responses. Emotions such as Happy and Angry show slightly higher MAE values yet still maintain relatively high cosine similarity scores (0.871 and 0.763), reflecting consistent structural agreement even when absolute deviations increase.

Overall, the results show that the late fusion strategy enhances emotion regression accuracy, confirming the advantage of combining complementary physiological cues in this context.

#### 6.2.3. Comparison of Method II to Method I via Dominant-Emotion Mapping

To allow a direct comparison between the regression-based Method II and the categorical Method I, the continuous emotion intensity outputs of the test dataset of Method II were converted into dominant-emotion labels by selecting, for each frame, the class with the highest predicted percentage. This procedure enables Method II to be evaluated using the same categorical metrics and confusion-matrix structure as Method I.

Table 6 summarizes the multilabel classification performance of the late fusion model on the test set after converting Method II’s percentage-based outputs into dominant-emotion labels. The overall accuracy and micro-averaged F1-score of 0.78 show that the model performs well when evaluated over all samples, and the low Hamming loss (0.21) indicates that most frame-level predictions match the ground truth. However, the macro-averaged scores and balanced accuracy (0.49 and 0.48, respectively) reveal substantial variability across emotion categories, with minority classes receiving lower per-class performance.

Compared to Method I (Table 3), Method II achieves notably higher overall accuracy (0.78 vs. 0.68) and micro-averaged F1, demonstrating that the percentage-based modeling captures richer emotional information that translates into more accurate dominant-emotion predictions. At the same time, Method I achieves a substantially higher balanced accuracy (0.64 vs. 0.48) and higher macro recall, indicating more uniform behavior across categories. This may be due to the fact that Method I is trained exclusively on dominant-emotion labels, which provide a clearer and less noisy supervisory signal. In contrast, Method II incorporates small, low-intensity emotional cues in its percentage-based targets. While this allows Method II to model subtle affective variations, it also introduces additional ambiguity, leading to more frequent misclassifications, particularly for minority or weakly expressed emotions.

Overall, the comparison shows that Method II excels at capturing global patterns and frequent emotions, benefiting from its continuous representation, while Method I remains superior in class-balanced performance. The two approaches are therefore complementary: Method I provides more stable per-class behavior, whereas Method II offers stronger overall predictive accuracy when its continuous outputs are projected onto categorical labels.

Figure 7 presents the resulting confusion matrix, illustrating how Method II behaves when forced into a discrete decision framework. The model achieves strong recognition of several emotions, most notably *happy* (86.2%), *sad* (70.6%), and *scared* (75.4%), which aligns with the lower MSE values observed for these emotions during continuous prediction. In contrast, minority classes such as *surprised* and *disgusted* show higher confusion, particularly with *happy*, reflecting the inherent difficulty of predicting low-intensity or ambiguous expressions.

A particularly striking result is the complete absence of correct predictions for *angry* (0%), which is almost entirely confused with *happy*. This confusion is plausible, as both emotions share high-arousal characteristics [39] and may appear similar when expressed with low intensity. Moreover, children with autism often display atypical or ambiguous facial expressions [40], which can increase the difficulty of distinguishing high-arousal negative affect from more positively valenced excitement when cues are subtle or atypical. As a result, Method II struggles to separate these two categories when forced into a discrete decision framework.

Despite being trained with percentage-based labels rather than categorical targets, Method II performs competitively when mapped to dominant classes and, for several emotions, even exceeds the discrete classification performance of Method I. This demonstrates that continuous intensity-based modeling captures richer emotional information that can be effectively translated into categorical decisions. At the same time, Method I remains more balanced across classes, as it is optimized directly for discrete emotion boundaries.

Overall, the comparison highlights that Method II provides a more expressive representation of emotional states, while Method I yields more uniformly calibrated class decisions. The two methods are therefore complementary: Method I excels in categorical discrimination, whereas Method II better captures graded or mixed emotional expressions.

### 6.3. Comparison of Reported Performance

Table 7 presents a compact comparison of emotion recognition performances across unimodal and multimodal approaches, covering EDA-, HR/HRV-, TEMP-, and face-based methods, as well as fusion strategies. Although emotion recognition has been extensively studied in neurotypical adults, relatively few works focus specifically on children with autism spectrum disorder (ASD). The majority of existing systems have been developed and validated on adult laboratory datasets (such as MAHNOB or WESAD), which exhibit stable physiological patterns, clearly expressed emotions, and balanced label distributions.

In the unimodal category, EDA-based methods show variable outcomes: Ref. [12] reported 86% accuracy on MAHNOB using CNN-based ResNet, whereas [16] achieved 84% accuracy using an in-house dataset. Similarly, HR-based methods demonstrate moderate to strong performance, ranging from over 80% [13], while heart rate approaches on children yield substantially lower accuracy (39.8%) [17]. Temperature signals, though investigated less frequently, exhibit weak discriminative power, with 36.2–42.5% accuracy reported by [19].

The few multimodal studies that include autistic children tend to show improved robustness. Ref. [41] combined EEG, HR, and GSR in an in-house ASD dataset and achieved high performance (94.2%), highlighting the benefits of integrating complementary physiological information. In neurotypical adult datasets, multimodal fusion reliably outperforms unimodal baselines as well, as demonstrated by [15], who achieved 99.8% accuracy using combined BVP and EDA signals on WESAD. TEMP + HRV combinations also reached high performance when tested on adults, reaching 95.58% in [20]. A particularly relevant comparison is [14], which also targets emotion recognition in an ASD population. Their multimodal speech–face model achieved 72.40% accuracy on the ASD-Affect dataset, lower than typical results in neurotypical datasets and therefore consistent with the challenges observed in our work. Their task was somewhat less complex, as it involved only three broad classes (positive, neutral, negative), yet the accuracy remained modest, further underscoring the difficulty of emotion recognition in autistic children.

Compared with these models, the results obtained in this article reflect the substantially higher complexity of emotion recognition in autistic children. Method I (Face + EDA + HR + TEMP) achieved 67.92% accuracy on EMBOA. Although this value is lower than multimodal results reported, it remains reasonable considering the challenges of the EMBOA recordings. The dataset suffers from severe class imbalance, with certain emotions appearing far more frequently than others (e.g., disgust or angry). Emotional expressions in autistic children are often less exaggerated, more idiosyncratic, and may include masking or atypical facial behavior, further reducing the discriminative power of vision-based models. The annotation process is also demanding: EMBOA labels were provided independently by three human raters, and inter-rater agreement is naturally lower for ASD populations because emotions can be blended, ambiguous, or expressed through subtle physiological changes rather than facial cues alone.

Method II (Face + EDA + HR + TEMP), which was evaluated using percentage-based annotations, produced an MSE of 11.33. Its continuous, distribution-based predictions better reflect the non-discrete and often uncertain emotional expressions common in ASD. This approach mitigates inconsistencies between annotators and better captures the emotional tendencies rather than forcing a single categorical label.

Overall, the comparison indicates that although our models do not reach the high accuracies reported for datasets, their performance is competitive and realistic within the context of ASD-focused research. The lower values are primarily due to the strong class imbalance, subtle or atypical emotional expressions, and inherently challenging labeling conditions. These factors underscore both the importance and the difficulty of developing multimodal affect recognition systems tailored to autistic children while highlighting the potential advantages of percentage-based modeling for capturing emotional nuance in this population.

## 7. Conclusions and Discussions

This study investigated multimodal emotion recognition in autistic children using two complementary modeling approaches: Method I, based on categorical labels, and Method II, using percentage-based continuous emotion intensities. Across both approaches, our results provide several key insights into the relative contributions of physiological and visual modalities and the effectiveness of the late fusion strategy.

Unimodal models demonstrated substantial variability in predictive performance. Among physiological signals, electro dermal activity (EDA) consistently provided the most informative cues, achieving the lowest mean squared error (MSE) in continuous prediction and the highest accuracy in categorical recognition. Heart rate (HR) and temperature offered moderate predictive power, whereas facial expressions (FaceReader) were less reliable, particularly for minority emotions. These findings align with the previous literature highlighting the challenges of facial emotion recognition in autistic populations, where subtle or atypical emotion reduces model accuracy.

The limitations of unimodal methods underscore the importance of mutimodal fusion. Both Method I and Method II demonstrated that combining signals substantially improves performance. For categorical recognition, late fusion enhanced balanced accuracy and macro-F1 scores, mitigating strong biases observed in unimodal classifiers. Similarly, for percentage-based predictions, late fusion across physiological modalities reduced overall MSE and MAE while preserving high cosine similarity with ground-truth emotional patterns. These results confirm that physiological signals carry complementary information that can compensate for weaker visual cues, particularly in populations with atypical facial expressivity.

Despite these promising results, several limitations should be noted. First, the dataset size is modest, and class imbalance remains a challenge, particularly for less frequent emotions. Second, the methods do not explicitly handle “unknown” or “other” emotional states, which may limit applicability in unconstrained real-world settings. Third, while physiological signals provide valuable cues, sensor placement and noise susceptibility may affect generalization.

Future works could explore incorporating additional behavioral modalities such as speech or gaze and applying attention-based or transformer architecture. Investigating alternative fusion strategies, including early fusion or hybrid approaches, could provide complementary insights to late fusion. Expanding the dataset and testing across diverse ASD populations would strengthen the applicability of the proposed framework. Another valuable field of development could be addressing the problem of small sample size (in the case of model evaluation) using approaches like “leave one out”. Particularly, it would be worth considering having multiple experiments with single test sample.

Overall, this study illustrates that even a relatively simple late fusion model can provide meaningful insights into the integration of synchronized physiological and behavioral signals for emotion recognition. While the model is not yet highly powerful, it serves as an initial step toward developing more adaptive and robust affective computing systems for children with ASD, laying the groundwork for future research on multimodal recognition under challenging conditions.

## Figures and Tables

**Figure 1 sensors-25-07485-f001:**
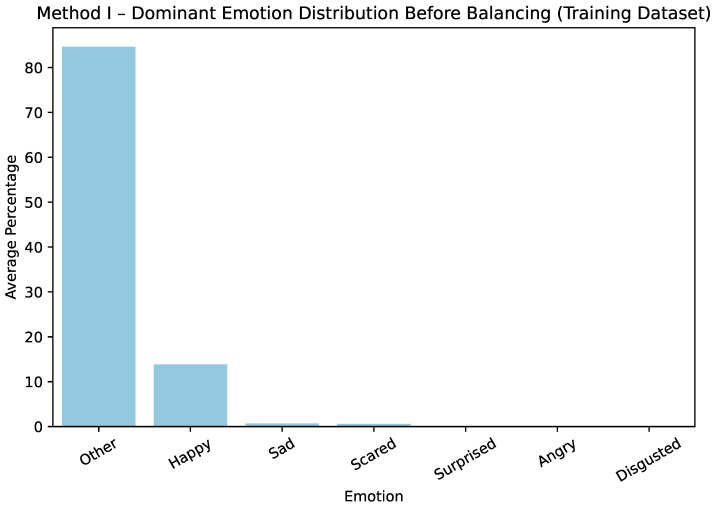
Method I—average dominant-emotion label distribution before balancing.

**Figure 2 sensors-25-07485-f002:**
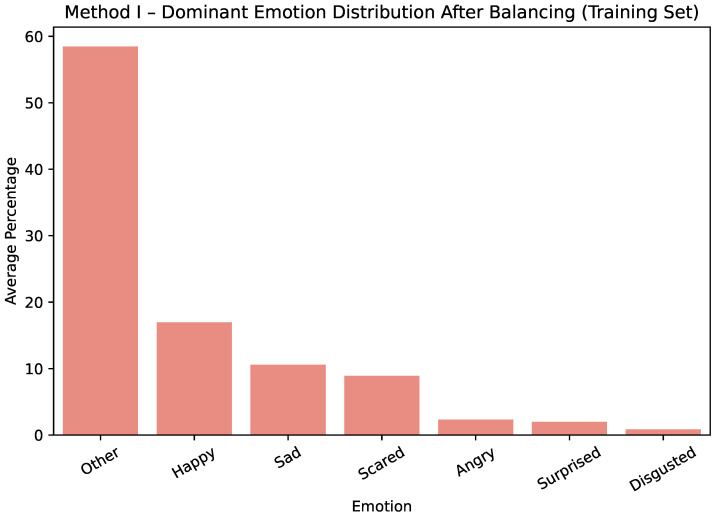
Method I—average dominant-emotion label distribution after balancing.

**Figure 3 sensors-25-07485-f003:**
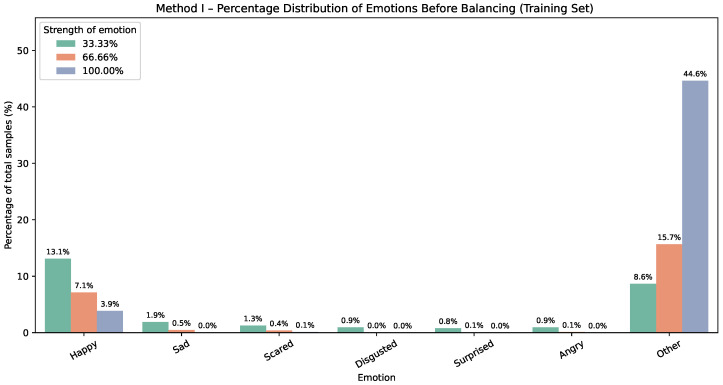
Method II—emotion distribution before balancing. Bars show percentages of the whole dataset; for each emotion, three bars indicate intensity levels (≈33.33%,66.66%,100%).

**Figure 4 sensors-25-07485-f004:**
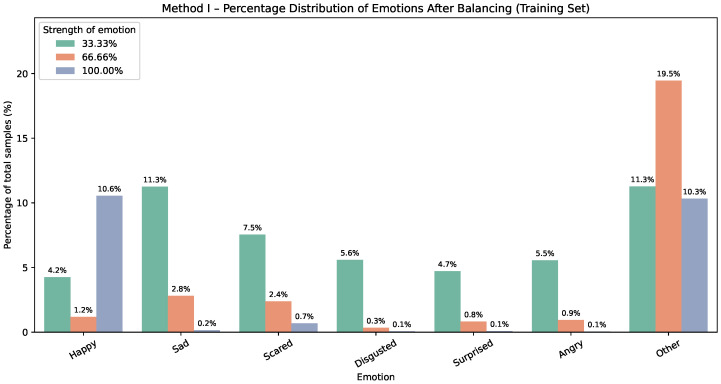
Method II—emotion distribution after balancing. Bars show percentages of the whole dataset; for each emotion, three bars indicate intensity levels (≈33.33%,66.66%,100%).

**Figure 5 sensors-25-07485-f005:**
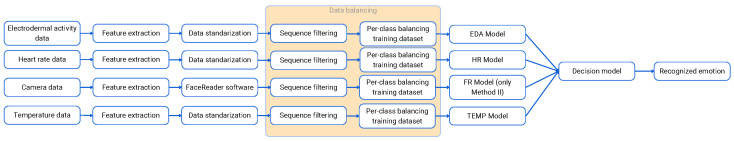
Overall architecture of the proposed emotion recognition framework. The system integrates preprocessing, unimodal models for facial expressions, EDA, HR, and TEMP, and a late fusion stage that outputs both categorical and distributional emotion representations.

**Figure 6 sensors-25-07485-f006:**
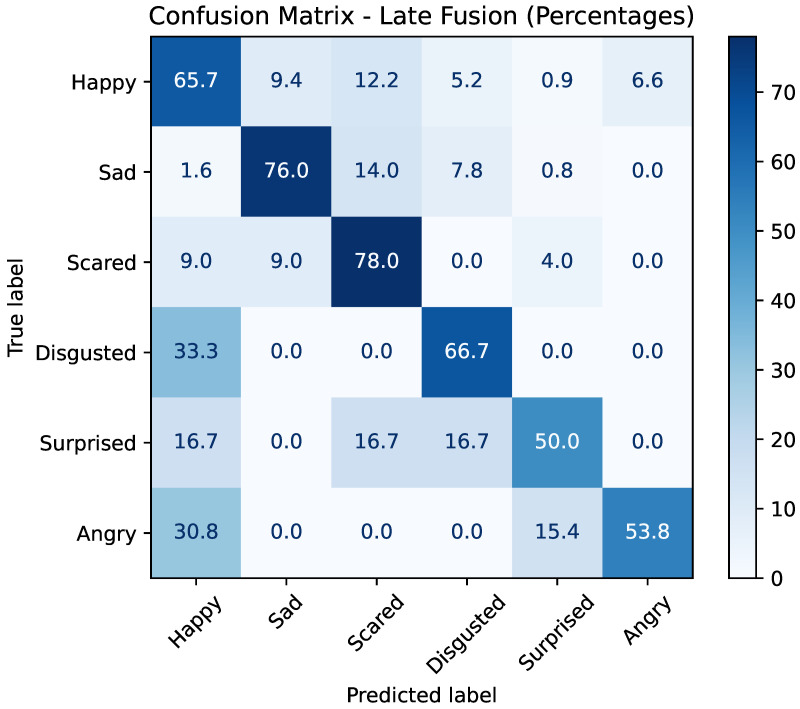
Confusion matrix of the late fusion model (Method I) on the EMBOA dataset. Each cell indicates the percentage of samples from a given true emotion class predicted as each output class.

**Figure 7 sensors-25-07485-f007:**
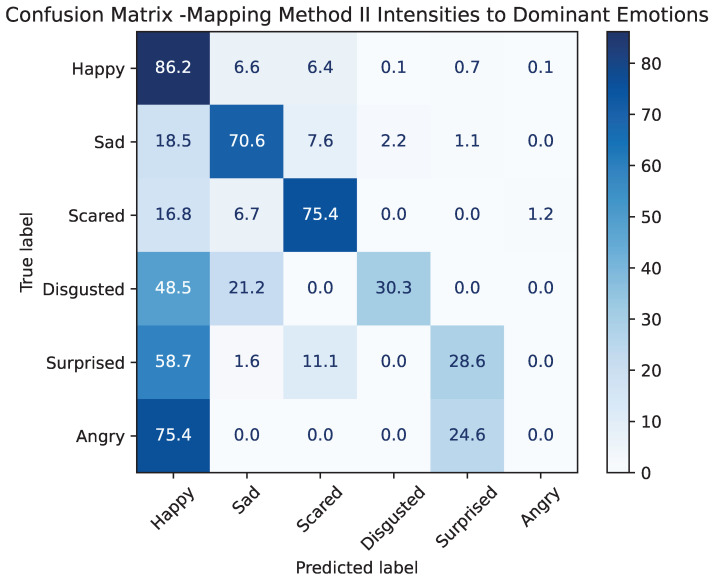
Confusion matrix obtained by mapping Method II emotion intensities to dominant categorical labels. Values represent the percentage of samples predicted for each true emotion class.

**Table 2 sensors-25-07485-t002:** Performance of unimodal models on the EMBOA dataset using categorical labels (Method I). Metrics include Accuracy, Balanced Accuracy, and Macro-F1.

Modality	Accuracy	Balanced Accuracy	Macro-F1
EDA	0.7057	0.2575	0.2486
HR	0.4926	0.1629	0.1438
TEMP	0.5148	0.2643	0.1879
FaceReader	0.4028	0.1846	0.1298

**Table 3 sensors-25-07485-t003:** Multilabel classification metrics for Method I.

Metric	Value
Accuracy	0.6792
Balanced accuracy	0.6438
F1-score (micro)	0.6792
Precision (macro)	0.4908
Recall (macro)	0.6438
F1-score (macro)	0.5101
Hamming loss	0.3208

**Table 4 sensors-25-07485-t004:** MSE results for each emotion across unimodal models for Method II.

Emotion	MSE			
	EDA Model	HR Model	Temp Model	FR Model
Happy	0.323	0.365	0.354	0.354
Sad	0.085	0.095	0.088	0.102
Scared	0.097	0.135	0.119	0.149
Disgusted	0.082	0.082	0.081	0.092
Surprised	0.103	0.099	0.098	0.104
Angry	0.107	0.105	0.108	0.112
Overall	0.133	0.147	0.141	0.152

**Table 5 sensors-25-07485-t005:** Evaluation per class (Method II).

Emotion	MSE	MAE	Cosine Similarity
Happy	0.249	0.435	0.871
Sad	0.086	0.240	0.804
Scared	0.076	0.224	0.868
Disgusted	0.080	0.276	0.789
Surprised	0.097	0.295	0.690
Angry	0.091	0.276	0.763
Overall	0.1133	0.2910	0.798

**Table 6 sensors-25-07485-t006:** Multilabel classification metrics obtained after mapping Method II percentage-based outputs to dominant-emotion labels.

Metric	Value
Accuracy	0.7846
Balanced Accuracy	0.4850
F1-score (micro)	0.7846
Precision (macro)	0.5030
Recall (macro)	0.4850
F1-score (macro)	0.4850
Hamming Loss	0.2153

**Table 7 sensors-25-07485-t007:** Comparison of emotion recognition performances.

Source	Modalities	Dataset	Metric	Value (%)	Emotions
[12]	EDA	MAHNOB	Accuracy	86 (CNN-based ResNet)	valence/arousal
[16]	EDA	In-house (Mindfield eSense)	Accuracy	84.0	non-active/active
[13]	HR	In-house	Accuracy	over 80%	happy, sad, neutral
[17]	HR	In-house (Children 8–11 years)	Accuracy	39.8 (SVM)	positive, negative, neutral
[19]	TEMP	In-house	Accuracy	36.2 and 42.5	joy, love, fear, sadness
[41]	EEG + HR + GSR	In-house (Autistic children)	Accuracy	94.2	happiness, anxiety, anger, sadness
[15]	BVP + EDA	WESAD	Accuracy	99.8	neutral, stress, amusement, meditation
[20]	TEMP + HRV	In-house	Accuracy	95.58	anger, disgust, fear, happiness, sadness, surprise
[14]	Speech + Face	ASD-Affect (from Kaur Bhat, 2019)	Accuracy	72.40	positive, neutral, negative
This work—Method I	Face + EDA + HR + TEMP	EMBOA	Accuracy	67.92	joy, sadness, anger, fear, surprise, disgust
This work—Method II	Face + EDA + HR + TEMP	EMBOA	MSE	11.33	joy, sadness, anger, fear, surprise, disgust

## Data Availability

The data used in this study were obtained from the third party [2] and are not publicly available.

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
