# Peer review of "Late Fusion Model for Emotion Recognition from Facial Expressions and Biosignals in a Dataset of Children with Autism Spectrum Disorder"

_sensors, 2025, doi:10.3390/s25247485_

Round 1

Reviewer 1 Report

Comments and Suggestions for Authors

This paper presents a multimodal fusion model for emotion recognition for children with autism spectrum disorder which is challenge and important. Facial expressions and bio-signals are fused at the decision level to recognize the emotion.  Two methods are designed to fulfil the task.  This paper is well written and easy to follow. One several minor points:

  1. These are many texts to describe the dataset, which is not collected in the presented work. Suggest to shortening the texts to let reader focus more on the main theme of this paper.
  2. Figures in this manuscript are very blur even though it is large enough, suggest to provide figures with high-resolution.
  3. Please check carefully throughout the paper to correct the typos, such as ‘iven’ in the beginning of section 5.1.

Author Response

Dear Reviewer, 

Thank you very much for your comments. We have analised them carefully and in the following sections we have responded to them in detail. 

Comment 1 

These are many texts to describe the dataset, which is not collected in the presented work. Suggest to shortening the texts to let reader focus more on the main theme of this paper. 

Response 1 

Thank you very much for your valuable suggestion. We greatly appreciate your opinion regarding the dataset description. We have retained a more detailed description of the dataset in the manuscript to make it self-contained and avoid requiring readers to consult external references. While we understand that some readers may prefer a shorter summary, we found that providing additional detail often helps clarify the dataset’s characteristics. We hope that this more detailed description will provide useful context without detracting from the main focus of the paper, and we sincerely appreciate your understanding of this design choice. 

Comment 2 

Figures in this manuscript are very blur even though it is large enough, suggest to provide figures with high-resolution. 

Response 2 

Thank you for your comment. We have addressed this by generating new plots using a vector graphics format, which ensures higher resolution and clarity. The figures in the revised manuscript now have improved quality and are easier to read. 

Comment 3 

Please check carefully throughout the paper to correct the typos, such as ‘iven’ in the beginning of section 5.1. 

Response 3 

We thank the Reviewer for carefully identifying these issues. The typographical error at the beginning of Section 5.1 has been corrected-the sentence now properly begins with “Given that physiological signals…”.  The small typing corrections have now been incorporated into the revised version. We also performed additional proofreading. 

Reviewer 2 Report

Comments and Suggestions for Authors

The reported results lack internal consistency. A balanced accuracy of 78% suggests a reasonably performing classifier, while an MSE of 2.3% (RMSE ≈ 0.15) for likelihood estimation indicates a model with poorly calibrated and uncertain probability outputs. These two results are incompatible when solving such a complex real-world task as emotion recognition in children with ASD.

Other questions, comments and remarks.

Line 201: "happy, sad, scared, disgusted, surprised, and angry". The literature and existing datasets suggest most common emotional state is neutral, how does it match to the presented approach "because FaceReader records only at change points, first the event log was converted to a regular series by sampling the closest preceding label at 1 Hz (forward fill between change points), producing per-second categorical labels."? If the most common in practice emotional state is not accounted for and instead rounded up to the most recent state, the proportion of each accounted for state becomes questionable.

Line 264: "with Neutral and Happy dominating". Neutral emotion is not mentioned as available in Line 201.

Line 279: "To achieve this, the 1 Hz data were resampled at 4 Hz, producing four synthetic variants per sequence. The resampled sequences were stored separately to ensure reproducibility". Resampling a 1 Hz signal to 4 Hz does not create new information. It simply creates a smoother version of the exact same sequence by interpolating between the original data points. One 10-second sequence of a child looking disgusted, turns into four synthetic sequences that are all, for all practical purposes, the exact same event. Because the four synthetic variants are nearly identical copies of the original sequence, if they are distributed across training and test sets, the model will be tested on data it has almost already seen. This approach may introduce severe data leakage between the training and test sets. If any of these synthetic variants are placed in different splits, the model will be evaluated on data that is virtually identical to its training data, rendering any performance metrics (like the reported 78% balanced accuracy) irreproducible.

Line 296: "The framework employs a dual analysis of emotional states". In any standard machine learning setup for multi-class classification, Path 2 is simply a prerequisite for Path 1. Presenting them as separate analytical paths may create a false sense of methodological complexity. What presented here is doing one thing (estimating a distribution) and then deriving two results from it (the argmax and the distribution itself). Despite the language used ("dual analysis," "captures complexity and ambiguity," "graded emotional profiling"), the underlying method is a standard practice, any modern classifier that outputs probabilities (like a softmax layer in a neural network) is inherently performing this "dual analysis.". With this interpretation, the "78% balanced accuracy" might just be the performance of taking the argmax of the distribution and the "MSE of 2.3%" might simply measure the error in estimating that same distribution.

The "emotion distribution" is only as good as the labels it is trained on. If the dataset labels are noisy, subjective, or poorly capture blended states (which is highly likely with ASD emotion data), then the model distribution will just be a confident or unconfident reflection of those flawed labels. It doesn’t introduce a novel way to handle ambiguity it is just reporting the standard confidence scores that every probabilistic classifier produces.

Line 304, next three positions are present further: "A late fusion strategy was employed to integrate the multimodal data"

Line 306: "Preserves modality-specific strengths". While late fusion does train separate models, it often does so at the cost of overall performance. A powerful model in one modality can easily dominate the fusion step, or the final classifier can struggle to reconcile conflicting predictions from the unimodal models if it hasn't learned how they correlate.

Line 306: "Allows flexible integration without imposing premature assumptions about interaction". In affective computing and especially with ASD, the interaction between modalities is often the most informative signal. A late fusion model, which processes modalities in isolation, is fundamentally incapable of learning these crucial, synergistic relationships.

Line 308: "Benefits from both specialized processing and a holistic view". Late fusion is a baseline method, not a state-of-the-art approach. Presenting it as a strategic framework without comparing it to more modern alternatives (like the ones mentioned above) significantly weakens the paper's methodological rigor.

Line 327: missing the capital letter.

Line 391: "This layer improves the modeling of interdependencies between modalities and outputs the final multi-label emotion probabilities.". With the described approach: "unimodal predictions are calculated as a weighted sum and fed this single, fused vector into a small neural network.", the input to this layer is a weighted sum of the final predictions from the unimodal models. At this stage, the rich, low-level or high-level features that could reveal cross-modal correlations have already been compressed into a single, confident prediction per modality. The FC layer can only perform a non-linear calibration or re-weighting of this already-fused decision vector. It cannot model the underlying, feature-level interactions between the individual signals. To genuinely model interdependencies, an intermediate fusion strategy would be required, where features from the different modalities are combined before the final classification layers, allowing the model to learn joint representations.

Line 506: the conclusion section, see next comments.

Line 508: The interpretation that the improvements stem from the "complementary nature" of the signals is speculative and unsupported by the experimental design. The late fusion approach, which processes each modality in isolation, is fundamentally incapable of learning the complex, time-dependent interactions as described. Without ablation studies or a fusion method that can actually model cross-modal dynamics, these explanations appear as post-hoc rationalizations.

Line 511: The claim that the model detects "subtle and co-occurring emotional states" is critically undermined by two key methodological flaws. First, the "emotion distribution" is a standard classifier output, not a novel method for multi-label learning. Second, the reported performance might be accidentally inflated by the data leakage introduced through the resampling-based augmentation. The model apparent sensitivity may simply reflect overfitting to near-duplicate sequences rather than a genuine capacity to generalize.

Line 515: The paper struggles to demonstrate a meaningful advantage for late fusion. The comparison is only made against unimodal baselines, which is a very low bar. To validate the approach, a comparison against modern intermediate fusion methods is critical. Given that intermediate fusion is specifically designed to model the cross-modal interactions the authors discuss, it is highly likely that such methods would outperform the presented model.

Author Response

Dear Reviewer, 

Thank you very much for your comments. We have analised them carefully and in the following sections we have responded to them in detail. 

Comment 1 

The reported results lack internal consistency. A balanced accuracy of 78% suggests a reasonably performing classifier, while an of 2.3% (RMSE ≈ 0.15) for likelihood estimation indicates a model with poorly calibrated and uncertain probability outputs. These two results are incompatible when solving such a complex real-world task as emotion recognition in children with ASD. 

Response 1 

Thank you very much for highlighting this issue. We have analised the method in question carefully and identified an instance of data leakage. Therefore, we have corrected the data-augmentation mechanism so that it now operates strictly on the training set and does not involve any external data. These changes are described in Section 3.4.3, Data Balancing. We have then repeated the experiment and updated the results accordingly. What is more, the results we obtained here were used to derive the dominant emotion. This result was evaluated, and the accuracy, f-score and other metrics were calculated and compered with the respective results obtained via Method I. We reported this in a new subsection 6.2.3, “Comparison of Method II to Method I via Dominant-Emotion Mapping,” which clarifies the relationship between the two methods and provides a direct comparison. The revised results now report the accuracy for Method I of 68% vs 78% for Method II with overall MSE of 11%.  

Comment 2 

Line 201: "happy, sad, scared, disgusted, surprised, and angry". The literature and existing datasets suggest most common emotional state is neutral, how does it match to the presented approach "because FaceReader records only at change points, first the event log was converted to a regular series by sampling the closest preceding label at 1 Hz (forward fill between change points), producing per-second categorical labels."? If the most common in practice, emotional state is not accounted for and instead rounded up to the most recent state, the proportion of each accounted for state becomes questionable. 

Response 2 

Thank you for raising this important point. Although neutral is indeed frequently reported in the literature as the most common emotional state, the characteristics of the EMBOA dataset do not allow us to treat neutral as a separate, reliably identifiable emotion category. 

First, the BORIS annotations used to generate ground-truth labels include only the seven emotions (namely: contempt, happy, sad, scared, disgusted, surprised, and angry) and do not define a neutral state. When none of the emotions is selected for a given moment, it is not possible to determine whether the child is truly neutral or simply displaying an expression that does not match any of the specified emotions. For this reason, such moments cannot be confidently labeled as neutral and were therefore assigned to a general category we refer to as “other”. 

Second, in the FaceReader output, the category neutral is detected, but for consistency we treat it in the same way: if FaceReader identifies neutral, we interpret it as the absence of any of the target emotions, not as an additional emotion class. Thus, neutral detections are grouped into the “other” category. 

Regarding the remark about forward filling, we agree that the original statements could have been misleading. There is no rounding of neutral into another emotion. Instead, forward filling simply preserves the last available FaceReader label, and whenever this label is neutral, the corresponding 1 Hz frame is assigned to the “other” category. To avoid misunderstanding, we have reformulated section 3.4.1 and the first three paragraphs of section 3. 

Comment 3 

Line 264: "with Neutral and Happy dominating". Neutral emotion is not mentioned as available in Line 201. 

Response 3 

Thank you for pointing it out. In line 264 we used the term Neutral to describe the lack of any of the specified emotions, which could be misinterpreted as another emotion, which was not our intention. To avoid this misunderstanding, the revised version explicitly introduces the term “other” emotion, which was explained in detail in response to the comment 2.  We corrected line 264 (now 304) to omit confusion. The new version of the sentence is as follows:  

“The raw label distribution was highly skewed, with no emotion labeled Other and Happy dominating, and minority classes - particularly Disgusted, Angry, and Surprised - appeared only rarely.” 

Comment 4 

Line 279: "To achieve this, the 1 Hz data were resampled at 4 Hz, producing four synthetic variants per sequence. The resampled sequences were stored separately to ensure reproducibility". Resampling a 1 Hz signal to 4 Hz does not create new information. It simply creates a smoother version of the exact same sequence by interpolating between the original data points. One 10-second sequence of a child looking disgusted, turns into four synthetic sequences that are all, for all practical purposes, the exact same event. Because the four synthetic variants are nearly identical copies of the original sequence, if they are distributed across training and test sets, the model will be tested on data it has almost already seen. This approach may introduce severe data leakage between the training and test sets. If any of these synthetic variants are placed in different splits, the model will be evaluated on data that is virtually identical to its training data, rendering any performance metrics (like the reported 78% balanced accuracy) irreproducible. 

Response 4 

Thank you for bringing attention to this issue. We agree that naïvely resampling a 1 Hz signal to 4 Hz could create near-duplicate sequences and lead to data leakage if such synthetic samples were distributed across different dataset splits. 

In our case, only the HR signal is originally sampled at 1 Hz. Both EDA and TEMP are natively recorded at 4 Hz, so no interpolation or artificial generation is applied to these signals. For HR, the value is repeated across the four frames because no higher-frequency information exists, and this repetition is used solely to preserve temporal alignment with the 4 Hz physiological streams. 

Importantly, we have clarified in the manuscript that upsampled/augmented sequences are added exclusively to the training set. The validation and test sets remain completely untouched, meaning that no synthetic variants can leak into the evaluation splits. This prevents the possibility of training–test contamination and ensures that reported performance metrics remain reproducible and free from data leakage. As a result, the training and test sets remain fully disjoint, ensuring that the evaluation reflects the model’s performance on unseen data. This procedure guarantees that the results reported in the manuscript are reliable and unbiased. 

We have revised the text accordingly to avoid misunderstanding and to explicitly state that oversampling and sequence expansion are applied only within the training dataset. The changes are mainly introduced to per-class balancing of subsection 3.4.3 and 3.4.4. 

Comment 5 

Line 296: "The framework employs a dual analysis of emotional states". In any standard machine learning setup for multi-class classification, Path 2 is simply a prerequisite for Path 1. Presenting them as separate analytical paths may create a false sense of methodological complexity. What presented here is doing one thing (estimating a distribution) and then deriving two results from it (the argmax and the distribution itself). Despite the language used ("dual analysis," "captures complexity and ambiguity," "graded emotional profiling"), the underlying method is a standard practice, any modern classifier that outputs probabilities (like a softmax layer in a neural network) is inherently performing this "dual analysis.". With this interpretation, the "78% balanced accuracy" might just be the performance of taking the argmax of the distribution and the "MSE of 2.3%" might simply measure the error in estimating that same distribution. 

Response 5 

Thank you for raising this point. We agree that, in a standard multi-class classification setting, probability estimation and taking the argmax are closely related operations. However, in our work the two modeling paths (Method I and Method II) are not simply two outputs derived from a single probability-estimating model. 

First, Method I does not operate on percentage-based labels. It is trained directly on categorical majority-vote labels using a different architecture optimized for discrete classification. 
Second, Method II uses percentage-based labels and a regression-style neural network specifically designed to predict continuous emotion intensities. These two models differ in architecture, loss functions, target representations, and training dynamics. 

Nevertheless, following the reviewer’s insight, we extended Method II by also deriving dominant-emotion predictions (via argmax) and reporting accuracy for this secondary outcome. This allows a more rigorous comparison with Method I. This issue was previously mentioned in response to the comment 1. 

Comment 6 

The "emotion distribution" is only as good as the labels it is trained on. If the dataset labels are noisy, subjective, or poorly capture blended states (which is highly likely with ASD emotion data), then the model distribution will just be a confident or unconfident reflection of those flawed labels. It doesn’t introduce a novel way to handle ambiguity it is just reporting the standard confidence scores that every probabilistic classifier produces. 

Response 6 

Regarding the interpretation of the predicted “emotion distributions”: we fully agree that the quality of any distributional output is limited by the quality of the underlying labels. In ASD datasets where emotional expressions may be ambiguous, atypical, or inconsistently interpreted annotation uncertainty is an inherent challenge. Our goal here was not to propose a novel way of resolving or modeling ambiguity in emotion labels. Instead, our objective was to evaluate whether, given such challenging and noisy annotations, it is still possible to construct models that achieve meaningful performance metrics. 

To make this clear, we added additional sentences in the manuscript discussing the nature of the annotations, the subjectivity and uncertainty inherent in the labeling process, and the limitations this introduces. (line 212 to 216: “The annotation was carried out independently by three annotators (female students from Gdańsk University of Technology). Annotators based their decisions primarily on facial expressions and body movements, but they also considered the context of each scenario and, when understandable, elements of speech and vocal tone [3]”.)  We hope this added clarification helps to contextualize the intent of our dual-path framework: it is not meant to suggest methodological novelty in probability estimation, but rather to assess whether different labeling schemes—categorical and percentage-based enable any viable emotion recognition performance under the constraints of the EMBOA dataset. 

Comment 7 

Line 304, next three positions are present further: "A late fusion strategy was employed to integrate the multimodal data" 

Response 7 

Thank you for pointing it out. We meant “A late fusion strategy was employed to integrate the multimodal models’ outputs", and this version is in the revised version of the paper. We are sorry for the confusion caused. 

Comment 8 

Line 306: "Preserves modality-specific strengths". While late fusion does train separate models, it often does so at the cost of overall performance. A powerful model in one modality can easily dominate the fusion step, or the final classifier can struggle to reconcile conflicting predictions from the unimodal models if it hasn't learned how they correlate. 

Response 8 

Thank you for pointing it out. In our work we utilized the assumptions that each modality model has more capacity to recognize some specific emotions. Additionally, we have chosen the late fusion approach as a first to check because while each modality is processed independently, the system can often provide a useful prediction even in the absence of data from one modality. It is typical in EMBOA dataset. 

While late fusion allows each unimodal model to leverage modality-specific information, it may also introduce challenges. A dominant modality can outweigh weaker ones during fusion, and the final classifier may struggle when unimodal predictions conflict. In our case, however, the performance across modalities was balanced, and late fusion achieved complementary integration without major dominance effects. 

Comment 9 

Line 306: "Allows flexible integration without imposing premature assumptions about interaction". In affective computing and especially with ASD, the interaction between modalities is often the most informative signal. A late fusion model, which processes modalities in isolation, is fundamentally incapable of learning these crucial, synergistic relationships. 

Response 9 

Thank you for pointing it out. Late fusion indeed enables modular training and flexible integration of modalities, but it does not capture cross-modal dependencies that may be especially informative in affective computing and ASD research. Due to the inability to recognize the face in many moments in the recordings, the late fusion approach was our first choice. Still, we plan to conduct the research further and apply other fusions.  

Comment 10 

Line 308: "Benefits from both specialized processing and a holistic view". Late fusion is a baseline method, not a state-of-the-art approach. Presenting it as a strategic framework without comparing it to more modern alternatives (like the ones mentioned above) significantly weakens the paper's methodological rigor. 

Response 10 

Thank you for pointing it out. In this study, late fusion was employed as an interpretable baseline approach that combines the outputs of modality-specific models. While this strategy provides a straightforward means of integration, it does not represent a state-of-the-art multimodal fusion technique. Future work may explore more advanced fusion methods, such as attention-based or hybrid models, to capture richer inter-modality relationships. To clarify this, we have revised the manuscript to explicitly describe late fusion as a baseline approach and to acknowledge that more advanced fusion techniques, like the ones mentioned earlier, could further enhance performance in future work. In particular we rephrased subsection 2.5 and Conclusions and Discussions (section 7). 

Comment 11 

Line 327: missing the capital letter. 

Response 11 

Thank you for pointing it out. We corrected that. 

Comment 12 

Line 391: "This layer improves the modeling of interdependencies between modalities and outputs the final multi-label emotion probabilities.". With the described approach: "unimodal predictions are calculated as a weighted sum and fed this single, fused vector into a small neural network.", the input to this layer is a weighted sum of the final predictions from the unimodal models. At this stage, the rich, low-level or high-level features that could reveal cross-modal correlations have already been compressed into a single, confident prediction per modality. The FC layer can only perform a non-linear calibration or re-weighting of this already-fused decision vector. It cannot model the underlying, feature-level interactions between the individual signals. To genuinely model interdependencies, an intermediate fusion strategy would be required, where features from the different modalities are combined before the final classification layers, allowing the model to learn joint representations. 

Response 12 

Thank you for pointing it out. Truly, the fusion layer receives weighted unimodal prediction vectors and performs a non-linear combination to produce the final multi-label emotion probabilities. This stage calibrates and integrates the unimodal decisions but does not directly capture feature-level interdependencies between modalities. Future work will explore intermediate fusion strategies to enable joint representation learning across modalities. 

Comment 13 

Line 508: The interpretation that the improvements stem from the "complementary nature" of the signals is speculative and unsupported by the experimental design. The late fusion approach, which processes each modality in isolation, is fundamentally incapable of learning the complex, time-dependent interactions as described. Without ablation studies or a fusion method that can actually model cross-modal dynamics, these explanations appear as post-hoc rationalizations. 

Response 13 

Thank you very much for this insightful comment. We agree that the statements used could have been misinterpreted, exaggerating the model's capabilities. In the revised manuscript, we have removed the speculative claim that the performance gains result from the “complementary nature” of the signals. As you correctly pointed out, our late fusion approach processes each modality independently and cannot capture cross-modal or time-dependent interactions. We have rephrased the statements to clearly express that the improvements likely reflect the combined contribution of independently learned modality-specific features, rather than any learned inter-modal dynamics. 
We appreciate your careful reading and constructive feedback, which helped us ensure that our interpretations remain fully aligned with the implemented methodology and observed results. 

Comment 14 

Line 511: The claim that the model detects "subtle and co-occurring emotional states" is critically undermined by two key methodological flaws. First, the "emotion distribution" is a standard classifier output, not a novel method for multi-label learning. Second, the reported performance might be accidentally inflated by the data leakage introduced through the resampling-based augmentation. The model apparent sensitivity may simply reflect overfitting to near-duplicate sequences rather than a genuine capacity to generalize. 

Response 14 

Thank you very much for this insightful comment. Following your observations stated here and in Comment 1, we carefully re-examined our data augmentation procedure and indeed identified an instance of data leakage that could have influenced the reported performance. We have corrected this issue by ensuring that all augmentation is now applied strictly within the training set, with no overlap with the test data. All experiments have been re-run, and all results, figures, and corresponding sections of the manuscript have been updated accordingly. In the revised version, we no longer state that the model detects “subtle and co-occurring emotional states,” as this phrasing actually overstated the implications of the standard classifier output. Instead, we now clarify that the model produces a distribution over emotion classes, which reflects graded likelihoods rather than the detection of simultaneous emotional states. The updated results further show that the MAE observed in Method II arises from the characteristics of the continuous labels used in that approach, rather than from leakage. At the same time, auxiliary analyses such as cosine similarity demonstrate that the temporal patterns are captured less effectively than previously suggested, and this is now explicitly acknowledged in the manuscript. We are very grateful for your careful reading and for drawing attention to these issues, which has substantially improved both the methodological rigor and the clarity of the revised manuscript. 

Comment 15 

Line 515: The paper struggles to demonstrate a meaningful advantage for late fusion. The comparison is only made against unimodal baselines, which is a very low bar. To validate the approach, a comparison against modern intermediate fusion methods is critical.  

Given that intermediate fusion is specifically designed to model the cross-modal interactions the authors discuss, it is highly likely that such methods would outperform the presented model. 

Response 15 

Thank you for pointing this out. In the revised manuscript, we have expanded the discussion of fusion strategies and clarified the limitations of our comparative analysis. To the best of our knowledge, no intermediate fusion methods have been published for the EMBOA dataset or for emotion recognition in autistic children more broadly. Our review of the literature did not identify any prior work applying intermediate fusion in this specific context. We agree that such methods are explicitly designed to model cross-modal interactions and may indeed outperform late fusion approaches. For this reason, we now clearly state that the aim of the present study is not to establish the superiority of late fusion over more advanced fusion architectures, but rather to investigate whether a late fusion strategy can yield reasonable and interpretable results on the EMBOA dataset as an initial step. We also emphasize that exploring intermediate fusion techniques represents an important direction for future research and may offer a more comprehensive assessment of the benefits of cross-modal modeling. We thank you again for this thoughtful suggestion, which helped us refine both the scope and the framing of our study. 

Reviewer 3 Report

Comments and Suggestions for Authors

The paper presents result of an experimental study exploring effects of late feature fusion on performance of emotion recognition system. It is considering a challenging problem characterized by atypical emotion expression by children with ASD. Presented results are of interest to research community and could help overcome limitations of the existing unimodal approaches. In general, the paper is well written and organized, however, there are methodological questions that are very important for interpretation of presented results, which should be carefully addressed. In the following are some of the comments that could help resolve these potential issues and improve the manuscript content after necessary revision.

- Minor changes related to current text are typographical errors in the beginning of section 5.1 (missing “G”) and introduction of the abbreviation BVP in section 2.5 (abbreviation is used, but not introduced anywhere before in the text).

 - Major changes should address the question of experimental design, model training and evaluation.

Namely, although the text is mostly consistent and with enough details regarding the dataset setup and conducted experiments, there are some questions regarding their validity from methodological side. If these would be properly addressed in the text, given results could be regarded as valid. Thus, please carefully consider how to introduce necessary details and if necessary re-run the experiments in order to make them aligned with introduced evaluation details.

First, the choice to have two experimental tracks (one with hard labels – Method I, and the second one with soft labels – Method II) is fine and would be of interest to compare how it affects the model training and overall performance. But, for both types of methods, from the text it is not clear how the samples that were not labeled as belonging to any of 6 selected emotions are handled. Are these put in the class 7, denoting samples that were not assigned to any of the considered categories according to labeling methodology I, or they are discarded, as indicated later on in the text when the so called “zero vectors” are mentioned? Thus, are there 7 categories on which the models are trained, or only 6? And how exactly the “zero vectors” are formed? When 3 human labelers gave their answers, did they provide 7 numbers, which were averaged, in order to form soft labels for Method II (page 2, lines 43-44)? From the text it seems that 3 hard labels, provided by human examiners, were just averaged, to produce a “soft label” with predefined percentage of 33, 66, or 100%, as later used in Fig. 3, but this is not clearly stated in the text. Also, in case that this is correct interpretation, what is with cases when there was no intersection or consensus) among the 3 labelers’ (how these samples were labeled)?

In figure 1 there are only 6 categories, and we do not get information how many samples were labeled as class “other”.  Similarly, please try to add this information also related to Figure3.

On page 12, it is said that 6x3 =18, decisions are merged in order to provide input for the final decision fusion module based on regression classifier. But, one paragraph before, on page 11, it is described that EDA, HR, TEMP and Face Reader classification confidence scores are used as inputs for the late feature fusion. So it is 6x4 “features”, i.e. soft decisions from the previous stage, which are combined further in order to produce the final decision. Please comment on this and try to introduce necessary corrections. Also, please address the class other question (this is important for interpretation how the final system should be used – is it capable of providing decisions on 6 selected emotions, or it can also sustain from decision and label the inputs as uncategorized emotions? This can have certain implications on system application, as well on the interpretation of numerical results.

Hierarchical modeling approach with independent processing tracks for each modality is fine and could be effective as indicated by the previous related works, and presented numerical experiments. However, how the evaluation of the system with such design is performed is not clearly described in the text and should be better addressed in the revised version of the manuscript. Namely, in the text is said that for the purpose of resolving the class imbalance problem, additional stratified resampling of the initially collected dataset is performed. This is a very delicate operation in terms of later implications on the evaluation process, so please try to introduce more details, and if necessary re-run the experiments in order to reflect the following. If some of the samples that were collected at 1Hz are artificially resampled to resemble a 4Hz sequence, then these newly added synthetic samples share many properties with the original data. Although this is dome only for underrepresented categories, it is important that this process is carefully designed. In the text there is no information how this resampling operation is actually performed (are the measurements are just repeated, or some other type of data augmentation is performed?).

Also, after these newly added samples are added, Fig.2, how they are assigned to training and test sets later on in the experimental setup? It is a key question that directly influences the validity of the presented results. If the newly added samples are made by repeating the existing measurements, then they must not appear in the test set, if their copies were used for training. Thus, in the text is missing explanation how the training and test sets are formed, and what mechanisms were used to avoid mixing the copies of the same samples between these two sets that should always contain disjoint samples. Since there are different sequences, it could also be of interest to considered having samples from different sequences in training and test set, i.e. avoiding having test data from the same sequence from which the training data were extracted. Whatever design choice is made; it should be clearly described in the text. Thus, please try to add necessary details and make corresponding adaptations to the performed experiments.

In Fig. 6, the accuracy assessment matrix that is shown indicates that for some of the categories number of samples was relatively low, less than 2? Thus, please consider joining some of the classes, or introducing the category other, in order to be able to provide relevant results interpretation. The choice of presenting micro and macro F in given Tables is more than welcomed in case of imbalanced datasets.

In case of the final results corresponding to “Method II” labeling, could you please also provide the final hard decisions (derived from the given regression results), and compare performance metrics based on such hard labels, with the metrics computed using Method I? This would help directly compare the effect of having the “soft labels” approach against the classical setup with discrete labels.

Also, when training late fusion classifiers, and classifiers for each of input modalities at the first stage, are these two types of models trained jointly or separately? In the first case this should be mentioned in the text, while in the second it should be explained how the models were trained sequentially on the same training data (again it is important not to mix the samples on which the performance is measured, with the ones used for training). Please also address these questions.

On way to overcome the problem of small sample size (in case of model evaluation) could be to use leave one out approach. So please also consider having multiple experiments with single test sample. Such approach could be feasible if the training is not costly, which is also the information that is missing from the text (if possible, please provide some indicator of the training complexity/duration on hardware that was used).

For method II, it is said that the late fusion was performed with class re-weighting in order to tackle the class imbalance problem. How this mechanism, at the level of classifier is interacting with the previously performed stratified resampling that was made for the same purpose. Please comment on this design choice, and also add some comment to the text, if necessary.

I hope that you will manage to rewrite some parts of the text and perform necessary adaptations that will address these questions.

Nice work, good luck.

Author Response

Dear Reviewer, 

Thank you very much for your comments. We have analised them carefully and in the following sections we have responded to them in detail. 

Comment 1 

The paper presents result of an experimental study exploring effects of late feature fusion on performance of emotion recognition system. It is considering a challenging problem characterized by atypical emotion expression by children with ASD. Presented results are of interest to research community and could help overcome limitations of the existing unimodal approaches. In general, the paper is well written and organized, however, there are methodological questions that are very important for interpretation of presented results, which should be carefully addressed. In the following are some of the comments that could help resolve these potential issues and improve the manuscript content after necessary revision. 

Minor changes related to current text are typographical errors in the beginning of section 5.1 (missing “G”) and introduction of the abbreviation BVP in section 2.5 (abbreviation is used, but not introduced anywhere before in the text). 

Response 1 

We thank the Reviewer for carefully identifying these issues. The typographical error at the beginning of Section 5.1 has been corrected-the sentence now properly begins with “Given that physiological signals…”. In addition, the abbreviation blood volume pulse (BVP) has been introduced at its first occurrence in Section 2.5 to ensure clarity and consistency in the manuscript. Both corrections have now been incorporated into the revised version. We have additionally reviewed the manuscript for fixing similar issues. 

Comment 2 

Major changes should address the question of experimental design, model training and evaluation. 

Namely, although the text is mostly consistent and with enough details regarding the dataset setup and conducted experiments, there are some questions regarding their validity from methodological side. If these would be properly addressed in the text, given results could be regarded as valid. Thus, please carefully consider how to introduce necessary details and if necessary re-run the experiments in order to make them aligned with introduced evaluation details. 

First, the choice to have two experimental tracks (one with hard labels – Method I, and the second one with soft labels – Method II) is fine and would be of interest to compare how it affects the model training and overall performance. But, for both types of methods, from the text it is not clear how the samples that were not labeled as belonging to any of 6 selected emotions are handled.  

Are these put in the class 7, denoting samples that were not assigned to any of the considered categories according to labeling methodology I, or they are discarded, as indicated later on in the text when the so called “zero vectors” are mentioned? Thus, are there 7 categories on which the models are trained, or only 6? And how exactly the “zero vectors” are formed? When 3 human labelers gave their answers, did they provide 7 numbers, which were averaged, in order to form soft labels for Method II (page 2, lines 43-44)? From the text it seems that 3 hard labels, provided by human examiners, were just averaged, to produce a “soft label” with predefined percentage of 33, 66, or 100%, as later used in Fig. 3, but this is not clearly stated in the text. Also, in case that this is correct interpretation, what is with cases when there was no intersection or consensus) among the 3 labelers’ (how these samples were labeled)? 

Response 2. 

Thank you for your comment. We have compared the two methods, adding for the Method II the derived dominant emotion and calculating accuracy, f-score and other metrics. We reported this in the newly added Section 6.2.3, “Comparison of Method II to Method I via Dominant-Emotion Mapping”. 

With respect to your statement that from the text it is not clear how the samples that were not labeled as belonging to any of 6 selected emotions are handled, we explicitly added the “other” label. It was caused at first due to the fact that the BORIS annotations used to generate ground-truth labels include only the basic emotions. When none of the emotions is selected for a given moment, it is not possible to determine whether the child is neutral or simply displaying an expression that does not match any of the specified emotions. For this reason, such moments we refer to as “other”.  The emotion “contempt”, which was also annotated in BORIS (there were annotated seven, not six emotions), was excluded and marked as “other,” as it did not appear in the output from FaceReader. Second, in the FaceReader output, additionally to the basic emotions (6 not 7, without “contempt”) the category neutral is detected, but for consistency we treat it in the same way: if FaceReader identifies neutral, we interpret it as the absence of any of the target emotions, not as an additional emotion class. Thus, neutral detections are grouped into the “other” category. 

For Method I, hard labels were assigned using majority voting; if only one annotator marked an emotion while others did not, the sample was considered as “other”. For Method II, soft labels were obtained by averaging the annotators’ responses for each category. In cases where each annotator labelled the emotion differently, the resulting average produced a zero value (“zero vector”) for each category, which again corresponds to “other.” 

Samples labeled “other” were masked during training so that they did not contribute to the loss, while still preserving their sequence order. During evaluation, these “other” samples were excluded, as our primary goal was to assess the model’s ability to recognize the basic emotions. 

To avoid misunderstanding, a paragraph in the introduction section (section 1) was added and we have reformulated the first three paragraphs of section 3, where these aspects were explained in detail. 

Comment 3 

In figure 1 there are only 6 categories, and we do not get information how many samples were labeled as class “other”.  Similarly, please try to add this information also related to Figure3. 

Response 3 

Thank you for pointing it out. The information is added in figures number 1, 2, 3, 4 (both for Method I and Method II). 

Comment 4 

On page 12, it is said that 6x3 =18, decisions are merged in order to provide input for the final decision fusion module based on regression classifier. But, one paragraph before, on page 11, it is described that EDA, HR, TEMP and Face Reader classification confidence scores are used as inputs for the late feature fusion. So it is 6x4 “features”, i.e. soft decisions from the previous stage, which are combined further in order to produce the final decision. Please comment on this and try to introduce necessary corrections. Also, please address the class other question (this is important for interpretation how the final system should be used – is it capable of providing decisions on 6 selected emotions, or it can also sustain from decision and label the inputs as uncategorized emotions? This can have certain implications on system application, as well on the interpretation of numerical results. 

Response 4 

Thank you for pointing this out. We apologize for the typographical error. In the final version, the model uses four modalities—FaceReader, EDA, HR, and temperature—resulting in 6 × 4 features (soft decisions from the previous stage) that are combined in the final decision fusion module using a regression classifier. 

Regarding the “other” or unlabeled samples, these were not used in training or evaluation, as our goal was to recognize the six selected emotions. However, to preserve the temporal structure of the sequences, these samples were included in the dataset during training as masked entries, so they did not contribute to the loss but maintained sequence continuity. During evaluation, these “other” samples were excluded, ensuring that the system produces decisions only for the six target emotions. This clarifies that the final system does not assign labels to uncategorized emotions and is intended solely for the selected emotion categories. 

Comment 5 

Hierarchical modeling approach with independent processing tracks for each modality is fine and could be effective as indicated by the previous related works, and presented numerical experiments. However, how the evaluation of the system with such design is performed is not clearly described in the text and should be better addressed in the revised version of the manuscript. Namely, in the text is said that for the purpose of resolving the class imbalance problem, additional stratified resampling of the initially collected dataset is performed. This is a very delicate operation in terms of later implications on the evaluation process, so please try to introduce more details, and if necessary re-run the experiments in order to reflect the following. If some of the samples that were collected at 1Hz are artificially resampled to resemble a 4Hz sequence, then these newly added synthetic samples share many properties with the original data. Although this is dome only for underrepresented categories, it is important that this process is carefully designed. In the text there is no information how this resampling operation is actually performed (are the measurements are just repeated, or some other type of data augmentation is performed?). 

Also, after these newly added samples are added, Fig.2, how they are assigned to training and test sets later on in the experimental setup? It is a key question that directly influences the validity of the presented results. If the newly added samples are made by repeating the existing measurements, then they must not appear in the test set, if their copies were used for training. Thus, in the text is missing explanation how the training and test sets are formed, and what mechanisms were used to avoid mixing the copies of the same samples between these two sets that should always contain disjoint samples. Since there are different sequences, it could also be of interest to considered having samples from different sequences in training and test set, i.e. avoiding having test data from the same sequence from which the training data were extracted. Whatever design choice is made; it should be clearly described in the text. Thus, please try to add necessary details and make corresponding adaptations to the performed experiments. 

Response 5 

Thank you for bringing attention to this issue. We agree that naïvely resampling a 1 Hz signal to 4 Hz could create near-duplicate sequences and lead to data leakage if such synthetic samples were distributed across different dataset splits. 

In our case, only the HR signal is originally sampled at 1 Hz. Both EDA and TEMP are natively recorded at 4 Hz, so no interpolation or artificial generation is applied to these signals. For HR, the value is repeated across the four frames because no higher-frequency information exists, and this repetition is used solely to preserve temporal alignment with the 4 Hz physiological streams. 

Importantly, we have clarified in the manuscript that upsampled/augmented sequences are added exclusively to the training set. The validation and test sets remain completely untouched, meaning that no synthetic variants can leak into the evaluation splits. This prevents the possibility of training–test contamination and ensures that reported performance metrics remain reproducible and free from data leakage. As a result, the training and test sets remain fully disjoint, ensuring that the evaluation reflects the model’s performance on unseen data. This procedure guarantees that the results reported in the manuscript are reliable and unbiased.  

We have revised the text accordingly to avoid misunderstanding and to explicitly state that oversampling and sequence expansion are applied only within the training dataset. The changes are mainly introduced to per-class balancing of subsection 3.4.3 and 3.4.4. 

Comment 6 

In Fig. 6, the accuracy assessment matrix that is shown indicates that for some of the categories number of samples was relatively low, less than 2? Thus, please consider joining some of the classes, or introducing the category other, in order to be able to provide relevant results interpretation. The choice of presenting micro and macro F in given Tables is more than welcomed in case of imbalanced datasets. 

Response 6 

Thank you for pointing this out. We have revised the manuscript to improve the interpretability of the results under class imbalance. Specifically, we now report macro-F1 scores for each modality (EDA, TEMP, HR) as well as the macro-F1 of the late-fusion model, as shown in Table 3. We also reference balanced accuracy to provide a more reliable performance measure for imbalanced data. 

In addition, Figure 6 has been updated to present the confusion matrix in percentages rather than raw counts, which allows for a clearer comparison across classes with different sample sizes. Unfortunately, some emotion categories contain very few samples in the original dataset, and we intentionally generated only partially synthetic samples in the training set, as stated in previous comment, not to risk distorting the model correctness. 

Comment 7 

In case of the final results corresponding to “Method II” labeling, could you please also provide the final hard decisions (derived from the given regression results), and compare performance metrics based on such hard labels, with the metrics computed using Method I? This would help directly compare the effect of having the “soft labels” approach against the classical setup with discrete labels. 

Response 7 

Thank you for this valuable suggestion. In the revised manuscript, we added a new section titled “6.2.3. Comparison of Method II to Method I via Dominant-Emotion Mapping,” where we convert the continuous percentage-based outputs of Method II into hard dominant-emotion labels and directly compare them with the categorical predictions of Method I. This enables a one-to-one comparison between the two approaches under an identical evaluation framework. 

It is important to emphasize that the methods are supervised by fundamentally different annotation schemes: Method I uses majority voting, where samples with insufficient agreement are labelled as “other”, whereas Method II models the mean intensity across annotators, preserving subtle or low-intensity emotional cues. This issue is discussed in detail in response to the comment 2. 

Comment 8 

Also, when training late fusion classifiers, and classifiers for each of input modalities at the first stage, are these two types of models trained jointly or separately? In the first case this should be mentioned in the text, while in the second it should be explained how the models were trained sequentially on the same training data (again it is important not to mix the samples on which the performance is measured, with the ones used for training). Please also address these questions. 

Response 8 

Thank you for pointing this out. We have clarified the training procedure in the revised manuscript, and the description has been added to Section 5.2. Model training and settings. The unimodal models and the late-fusion model were trained separately for both methods and sequentially, not jointly. To be more precise, the training of each of the two unimodal models for the same modality (Method I and II) was done separately. 

The dataset was first split into training, validation, and test sets in a 60/20/20 ratio, and this split was fixed for all experiments. Only the 60% training split was balanced to address class imbalance; the validation and test sets remained untouched. This ensured that model evaluation was always performed on naturally distributed, unseen data. 

Each unimodal model (EDA, HR, TEMP, FaceReader) was then trained only on the balanced training portion of its modality, using the validation set exclusively for hyperparameter tuning. The test set was never used during training or tuning. 

After the unimodal models were trained independently, their predictions on the training split were used as inputs for training the late-fusion classifier. Importantly, the late-fusion model was trained solely on unimodal outputs derived from the training set, ensuring no leakage of validation or test information. During evaluation, the late-fusion classifier received unimodal predictions generated strictly from the test set. 

This sequential but clearly separated workflow guarantees that (1) balancing affects only the training process, and (2) performance metrics are computed exclusively on unseen data. These details have been added to the manuscript for clarity. 

Comment 9 

One way to overcome the problem of small sample size (in case of model evaluation) could be to use leave one out approach. So please also consider having multiple experiments with single test sample. Such approach could be feasible if the training is not costly, which is also the information that is missing from the text (if possible, please provide some indicator of the training complexity/duration on hardware that was used). 

Response 9 

Thank you for the valuable suggestion, we plan to utilize this approach in further studies. We also included this idea in the Conclusions and Discussions (section 7) of the manuscript to depict possible ways of network development. 

Comment 10 

For method II, it is said that the late fusion was performed with class re-weighting in order to tackle the class imbalance problem. How this mechanism, at the level of classifier is interacting with the previously performed stratified resampling that was made for the same purpose. Please comment on this design choice, and also add some comment to the text, if necessary. 

Response 10 

Thank you for pointing it out. Thank you for your valuable comment. We have taken it into account and updated the manuscript accordingly. In the revised version, the late-fusion model uses only stratified resampling to address class imbalance. We removed the previously mentioned class re-weighting mechanism, as it is not applied in the current implementation. Stratified resampling ensures balanced data distribution during training, making additional class-level weighting unnecessary. 

Round 2

Reviewer 2 Report

Comments and Suggestions for Authors

The text of the article has been significantly revised in accordance with the comments made by the reviewer.

The article can be accepted for publication.